# EXPLORING FEDERATED OPTIMIZATION BY REDUCING VARIANCE OF ADAPTIVE UNBIASED CLIENT SAMPLING

## ABSTRACT

Federated Learning (FL) systems usually sample a fraction of clients to conduct a training process. Notably, the variance of global estimates for updating the global model built on information from sampled clients is highly related to federated optimization quality. This paper explores a line of "free" adaptive client sampling techniques in federated learning without requiring additional local communication and computation. These methods could enhance federated optimization by providing global estimates for model updating with designed sampling probability. We capture a minor variant in the sampling procedure and improve the global estimation accordingly. Based on that, we propose a novel sampler called K-Vib, which solves an online convex optimization problem respecting federated client sampling tasks. It achieves improved a linear speed up on regret bound $\tilde{\mathcal{O}}\big(N^{\frac{1}{3}}T^{\frac{2}{3}}/K^{\frac{4}{3}}\big)$ with communication budget $K$. As a result, it significantly improves the performance of federated optimization. Theoretical improvements and intensive experiments on classic federated tasks demonstrate our findings.

## 1 INTRODUCTION

This paper studies the standard cross-device federated learning (FL) setting (Kairouz et al., 2021), which optimizes $x \in \mathcal{X} \subseteq \mathbb{R}^d$ to minimize a finite-sum objective:

$$\min_{x \in \mathcal{X}} f(x) := \sum_{i=1}^{N} \lambda_i f_i(x) := \sum_{i=1}^{N} \lambda_i \mathbb{E}_{\xi_i \sim \mathcal{D}_i}[F_i(x, \xi_i)], \tag{1}$$

where $N$ is the total number of clients, $\lambda_i \geq 0, \sum_{i=1}^{N} \lambda_i = 1$ is the weights of client objective. The $f_i : \mathbb{R}^d \to \mathbb{R}$ is a local loss function that depends on the local data distribution $D_i$ owned by client $i$ via $f_i(x) = \mathbb{E}_{\xi_i \sim \mathcal{D}_i}[F_i(x, \xi_i)]$ as its the stochastic results. Typically, federated optimization algorithm, e.g., FEDAVG (McMahan et al., 2017) that minimizes objective 1 basically follows a distributed learning protocol involving local and global update alternatively as shown in Algorithm 1.

In cross-device FL, communication and computation efficiency are the primary bottlenecks (Konečný et al., 2016; Yang et al., 2022), as the typical clients are mobile phones or IoT devices that have limited bandwidth and computation resources. Client sampling techniques are a feasible way to enhance federated learning efficiency (Wang et al., 2021), which are motivated by the fact that the data quality and quantity are in large variance across clients (Khan et al., 2021). Consequently, some clients can provide more informative updates than others in a communication round.

To fully utilize local information and enhance the training efficiency, a number of client sampling approaches have been proposed in the literature (Chen et al., 2020; Cho et al., 2020b; Balakrishnan et al., 2022; Wang et al., 2022; Malinovsky et al., 2023; Cho et al., 2023). Although they obtained promising results, most of these methods usually require additional communication or computation on the client side compared with vanilla FL protocol. However, some sampling techniques are not applicable in a resource-constrained FL system (Imteaj et al., 2021), where the devices have no additional computation and communication resources to conduct such a sampling task. Besides, sampling techniques also involve biasedness (Cho et al., 2020b; Wang et al., 2022) and unbiasedness (Borsos et al., 2018; El Hanchi & Stephens, 2020) as we concretely discussed in Appendix E.2.

Compared with biased sampling methods, unbiased client sampling methods are orthogonal with secure aggregation (Du & Atallah, 2001; Goryczka & Xiong, 2015; Bonawitz et al., 2017) and FL re-weighting algorithms that adjust $\lambda$ for fairness/robustness (Li et al.; 2021). Besides, unbiased client sampling methods maintain the optimization objective unbiasedness, which has been proved essential (Wang et al., 2020) to optimization quality.

---

**Algorithm 1** FedAvg with Unbiased Client Sampler

---

**Input:** Client set $S$, where $|S| = N$, client weights $\lambda$, times $T$, local steps $R$

1   Initialize sample distribution $p^0$ and model $w^0$

2   **for** *time $t \in [T]$* **do**

3      Server run sampling procedure to obtain sampled client set $S^t \sim p^t$

4      Server broadcast $x^t$ to sampled clients $i \in S^t$

5      **for** *each client $i \in S^t$ in parallel* **do**

6         $x_i^{t,0} = x^t$

7         **for** *local steps $r \in [R]$* **do**

8            $x_i^{t,r} = x_i^{t,r-1} - \eta_l \nabla F_i(x_i^{t,r-1}, \xi_i \sim \mathcal{D}_i)$ // `local SGD`

9         Client upload local updates $g_i^t = x_i^{t,0} - x_i^{t,R}$ to the server

10      Server builds global estimates $d^t = \sum_{i \in S^t} \lambda_i g_i^t / p_i^t$ and updates Model $w^{t+1} \leftarrow w^t - \eta_g d^t$

11      Server updates $p^{t+1}$ based on received information $\{\|g_i^t\|\}_{i \in S^t}$ // `adaptive`

---

In this paper, we build upon existing unbiased sampling methods in stochastic optimization literature (Salehi et al., 2017; Borsos et al., 2018; El Hanchi & Stephens, 2020), and focus on federated client sampling. Given the constraints of limited local communication and computation, our goal is to explore "free" client sampling techniques that leverage only the uploaded local update. It is expected to be powerful in improving federated optimization efficiency such as in edge-computing systems (Khan et al., 2021). To achieve this, we propose a novel adaptive client sampling method that aligns with the basic FL protocol outlined in Algorithm 1, with only modifications to the server sampling procedure and sampling distribution. By analyzing the optimal unbiased client sampling procedure and probability in Section 2, we extend adaptive unbiased sampling techniques using the independent sampling procedure. This procedure involves rolling dice independently for each client respecting a well-designed probability distribution. It also builds promising global estimates for global updates in FL. It achieves significant improvement in both theory and empirical experiments. Our contribution can be summarised as follows:

**Proposed novel sampler K-Vib.** To the best of our knowledge, we are the first work that extends the independent sampling procedure on adaptive client sampling in federated optimization. To find the best probability, we model the unbiased client sampling task in federated learning as an online convex optimization problem for gradient variance reduction. In this context, we theoretically proved that K-VIB achieves an expected regret bound $\tilde{\mathcal{O}}\big(N^{\frac{1}{3}} T^{\frac{2}{3}} / K^{\frac{4}{3}}\big)$ with a near-linear speed up, comparing with previous bound $\tilde{\mathcal{O}}\big(N^{\frac{1}{3}} T^{\frac{2}{3}}\big)$ (Borsos et al., 2018) and $\mathcal{O}\big(N^{\frac{1}{3}} T^{\frac{2}{3}}\big)$ (El Hanchi & Stephens, 2020), where $N$ is the number of clients, $K$ is communication budget (i.e., the expected number of sampled clients), $T$ is the communication rounds and $\tilde{\mathcal{O}}$ hides logarithmic factor.

**Enhancing federated optimization.** We principled the non-convex convergence analysis of classic federated optimization with arbitrary unbiased client sampling techniques in Theorem 3.9. The unique perspective connects the regret bound in online convex optimization with the convergence bound of federated optimization, which reveals the impacts of adaptive client sampling techniques in federated optimization. The result indicates our methods inherit the benefits of the online convex optimization task, and hence achieve faster convergence than baselines.

**Experiments evaluation.** We validate our theoretical findings using a Synthetic dataset and assess the performance of K-Vib in classic federated learning tasks. The results unequivocally demonstrate that the K-Vib sampler effectively constructs a robust sampling procedure and generates accurate global estimates by solving online convex optimization problems. The reduced variance in these global estimates accelerates the convergence of federated optimization, leading to faster model updates.

## 2 PRELIMINARIES

In this section, we clarify the concepts of unbiased client sampling techniques in federated learning and demonstrate the optimal solution with given local updates at any communication round $t$ and budget $K$ (the number of clients expected to be sampled). Then, we formulate the online convex optimization problem to obtain promising sampling probabilities and global estimates.

### 2.1 UNBIASED CLIENT SAMPLING AND ITS OPTIMALITY

Unbiased client sampling is defined with the global estimates $d^t$ and the global objective in 1. Letting communication budget $K$ as the expected number of sampled clients for each round, we give the variance of estimates in Line 10, Algorithm 1, respecting sampling probability $p^t$:

**Definition 2.1** (The global estimator $d^t$ variance). The sampling quality of arbitrary client sampling is related to the variance of its estimation, which can be formalized as:

$$\mathbb{V}(S^t \sim p^t; \lambda, g^t) := \mathbb{E}\left[\left\|d^t - \sum_{i=1}^{N} \lambda_i g_i^t\right\|^2\right] + \mathbb{E}\left[\left\|\sum_{i=1}^{N} \lambda_i g_i^t - \sum_{i=1}^{N} \lambda_i \nabla f_i(x^t)\right\|^2\right], \quad (2)$$

where $p_i^t > 0, \sum_{i=1}^{N} p_i^t = K = \mathbb{E}[|S^t|]$ for all $i \in [N], t \in [T]$, and $S \sim p^t$ denotes the sampling procedure used to create the sampling $S^t$. *To be consistent, the sampling probability $p$ always satisfies the constraint $p_i^t > 0, \sum_{i=1}^{N} p_i^t = K, \forall t \in [T]$ in this paper.*

We say a client sampling technique is unbiased if the sampling and estimates satisfy that $\mathbb{E}[d^t] = \sum_{i=1}^{N} \lambda_i g_i^t$, where the variance is given in the first term of 2. The second term represents the local drift induced by the multiple local SGD steps in federated optimization to save communication (McMahan et al., 2017). We involved the bias in our convergence analysis in Section 3.3. The optimality of the global estimator depends on the collaboration of sampling distribution $p^t$ and the corresponding procedure that outputs $S^t$. In detail, different sampling procedures associated with the sampling distribution $p$ build a different *probability matrix* $\mathbf{P} \in \mathbb{R}^{N \times N}$, which is defined by $\mathbf{P}_{ij} := \text{Prob}(\{i, j\} \subseteq S)$. Inspired by the arbitrary sampling (Horváth & Richtárik, 2019), we derive the optimal sampling procedure for the FL server in Lemma 2.2.

**Lemma 2.2** (Optimal sampling procedure). *For any communication round $t \in [T]$ in FL, noting the aforementioned notations including gradients $g_i^t$, sampling distribution $p^t$, weights $\lambda$, the variance of estimates in Equation 2 are related to $\mathbf{P}_{ij}$, which varies in a sampling procedure that creates $S^t$.*

*For random sampling[1] yielding the $\mathbf{P}_{ij} = Prob(i, j \in S) = K(K-1)/N(N-1)$, it admits:*

$$\mathbb{E}\left[\left\|\sum_{i \in S^t} \frac{\lambda_i}{p_i^t} g_i^t - \sum_{i=1}^{N} \lambda_i g_i^t\right\|^2\right] \leq \frac{N-K}{N-1} \sum_{i=1}^{N} \frac{\lambda_i^2 \|g_i^t\|^2}{p_i^t}. \quad (3)$$

*Analogously, for independent sampling[2] with $\mathbf{P}_{ij} = Prob(i, j \in S) = p_i p_j$, it admits:*

$$\mathbb{E}\left[\left\|\sum_{i \in S^t} \frac{\lambda_i}{p_i^t} g_i^t - \sum_{i=1}^{N} \lambda_i g_i^t\right\|^2\right] = \underbrace{\sum_{i=1}^{N} (1 - p_i^t) \frac{\lambda_i^2 \|g_i^t\|^2}{p_i^t}}_{\text{independent sampling}} \leq \underbrace{\frac{N-K}{N-1} \sum_{i=1}^{N} \frac{\lambda_i^2 \|g_i^t\|^2}{p_i^t}}_{\text{random sampling}}. \quad (4)$$

*In conclusion, the independent sampling procedure is the optimal sampling procedure that always minimizes the upper bound of variance regardless of the sampling probability.*

Directly utilizing the independent sampling procedure could obtain the variance reduction benefits, as we evaluated in Figure 5. We can enlarge the benefits via minimizing the variance in Equation 4 respecting probability $p$, which is generally given in Lemma 2.3.

---

[1]Random sampling procedure means that the server samples clients from a black box without replacement.
[2]Independent sampling procedure means that the server rolls a dice for every client independently to decide whether to include the client.

**Lemma 2.3** (Optimal sampling probability). *Generally, we can let $a_i = \lambda_i \|g_i^t\|, \forall i \in [N], t \in [T]$ for simplicty of notation. Assuming $0 < a_1 \leq a_2 \leq \cdots \leq a_N$ and $0 < K \leq N$, and $l$ is the largest integer for which $0 < K + l - N \leq \frac{\sum_{i=1}^{l} a_i}{a_l}$, we have*

$$p_i^* = \begin{cases} (K + l - N)\frac{a_i}{\sum_{j=1}^{l} a_j}, & \text{if } i \leq l, \\ 1, & \text{if } i > l, \end{cases} \tag{5}$$

*to be a solution to the optimization problem $\min_p \sum_{i=1}^{N} \frac{a_i^2}{p_i}$.*

**Remark.** Lemma 2.2 reveals the optimality of the sampling procedure of designing $S^t$ and Lemma 2.3 demonstrates the optimal probability distribution $p^t$. Despite the differences in methodology, previous importance sampling techniques in stochastic optimization (Salehi et al., 2017; Duchi et al., 2011; Boyd et al., 2004) and federated client sampling (Zhao et al., 2021; Borsos et al., 2018; El Hanchi & Stephens, 2020) designs sampling probability based on the sub-optimal gradient variance formulation in Equation 3. In this paper, we capture the minor variant in the sampling procedure as demonstrated in Lemma 2.2. And, we propose using the independent sampling procedure to enhance the power of the unbiased sampling technique. Motivated by the observation in Lemma 2.3, we explore an efficient adaptive sampling in the methodology section.

## 2.2 Adaptive Client Sampling as Online Convex Optimization

Directly computing Equation 5 in FL consumes tremendous device computation power for backprop-agation locally, as it requires the norm information of local update of ALL clients. However, it is not necessary to achieve the optimal sampling for federated optimization because of the data quality variance across clients. Instead, we can implement a sub-optimality sampling to obtain most of the benefits without requiring additional local computation and communication in FL.

To this end, we model the adaptive client sampling as an online convex optimization problem respecting sampling probability $p^t$ during federated optimization. Concretely, we denote the required feedback from clients as a function $\pi_t(i) := \|g_i^t\|$ and define the cost function $\ell_t(p) := \sum_{i=1}^{N} \frac{\pi_t(i)^2}{p_i}$ for online convex optimization[3]. Our target of building sampling probability is to minimize the *dynamic* regret between applied sampling probability and the Oracle:

$$\text{Regret}_D(T) = \frac{1}{N^2}\left(\sum_{t=1}^{T} \ell_t(p^t) - \sum_{t=1}^{T} \min_p \ell_t(p)\right), \tag{6}$$

where we set the client weight $\lambda_i = \frac{1}{N}, \forall i \in [N]$ as static for simply notation. $\text{Regret}_D(T)$ indicates the cumulative discrepancy of applied sampling probability and the *dynamic* optimal probability.

In this paper, we are to build an efficient sampler that outputs an exemplary sequence of independent sampling distributions $\{p^t\}_{t=1}^{T}$ such that $\lim_{T \to \infty} \text{Regret}_D(T)/T = 0$. Meanwhile, it enhances the quality of corresponding federated optimization algorithms.

## 3 Methodology of K-Vib Sampler

In this section, we introduce the design of the K-Vib sampler. Firstly, we identify a sub-optimal sampling probability and demonstrate, through theoretical analysis, that the deviation between this probability and the optimal one diminishes over time. Next, we present our method from two interconnected scenarios, offering a theoretical perspective. Lastly, we analyze the convergence of FedAvg, emphasizing the effects of unbiased client sampling techniques.

## 3.1 Sub-optimal Probability and its Vanishing Gap

Our methods rely on mild assumptions about local objective $f_i(\cdot), \forall i \in [N]$, and the convergence performance of applied optimization algorithms.

---

[3]Please distinguish the online cost function $\ell_t(\cdot)$ from local empirical loss of client $f_i(\cdot)$ and global loss function $f(\cdot)$. While $\ell_t(\cdot)$ is always convex, $f(\cdot)$ and $f_i(\cdot)$ can be non-convex.

**Assumption 3.1** (Local Convergence). *Let $\Pi_t := \frac{1}{N} \sum_{i=1}^{N} \pi_t(i)$ denote the average of feedback. Thus, we define $\pi_*(i) := \lim_{t \to \infty} \pi_t(i)$, $\Pi_* := \frac{1}{N} \sum_{i=1}^{N} \pi_*(i)$, $\forall i \in [N]$. Moreover, we could denote that $\frac{1}{T} \sum_{t=1}^{T} \Pi_t \geq \Pi_*$, $V_T(i) = \sum_{t=1}^{T} (\pi_t(i) - \pi_*(i))^2$, $\forall T \geq 1$, and $\pi_t(i) \leq G, \forall t \in [T], i \in [N]$.*

*A faster FL solver implies a lower bound for $|\pi_t(i) - \pi_*(i)|$, and hence $V_T(i)$. For instance, SGD roughly implements $|\pi_t(i) - \pi_*(i)| \leq \mathcal{O}(1/\sqrt{t})$, and hence implies $V_T(i) \leq \mathcal{O}(\log(T))$. Thus, the above theorem would translate into regret guarantees with respect to the ideal baseline, with an additional cost of $\tilde{\mathcal{O}}(\sqrt{T})$ in expectation.*

**Remark.** The Assumption 3.1 guarantees the sampling technique is applied in a converging federated optimization process. It indicates the sub-linear convergence speed of an optimization process, which commonly holds in non-convex optimization with SGD (Salehi et al., 2017; Duchi et al., 2011; Boyd et al., 2004) and federated optimization (Reddi et al., 2020; Wang et al., 2020; Li et al., 2019). Importantly, the $G$ denotes the largest feedback during the applied optimization process, instead of assuming bounded gradients. It can be justified by using differential privacy (Kairouz et al., 2021).

**Vanishing Hindsight Gap.** We decompose the original regret in Equation 6 as follows:

$$N^2 \cdot \text{Regret}_D(T) = \underbrace{\sum_{t=1}^{T} \ell_t(p^t) - \min_p \sum_{t=1}^{T} \ell_t(p)}_{\text{Regret}_S(T)} + \underbrace{\min_p \sum_{t=1}^{T} \ell_t(p) - \sum_{t=1}^{T} \min_p \ell_t(p)}_{\text{Hindsight Gap}}, \tag{7}$$

where the static regret $\text{Regret}_S(T)$ indicates the distance between a given sequence of probabilities and the best-*static* probability in hindsight; the second term indicates the gap between the best-*static* probability in hindsight and the ideal probabilities from the Oracle. Rely on the mild assumptions, we bound the second term of Equation 7 below:

**Theorem 3.2** (Vanishing Hindsight Gap). *Under Assumptions 3.1, sampling a batch of clients with an expected size of $K$, and for any $i \in [N]$ denote $V_T(i) = \sum_{t=1}^{T} (\pi_t(i) - \pi_*(i))^2 \leq \mathcal{O}(\log(T))$. For any $T \geq 1$, the averaged hindsight gap admits,*

$$\frac{1}{N^2} \left[ \min_p \sum_{t=1}^{T} \ell_t(p) - \sum_{t=1}^{T} \min_p \ell_t(p) \right] \leq \frac{2\sqrt{T}\Pi_*}{NK} \sum_{i=1}^{N} \sqrt{V_T(i)} + \left( \frac{1}{NK} \sum_{i=1}^{N} \sqrt{V_T(i)} \right)^2.$$

***Remark.*** Lemma 3.2 demonstrates the connection between the FL optimizer and the minimization of regret. That is, a fast convergence induces a lower bound of $V_t(i)$, yielding faster vanishing. As the hindsight gap vanishes with an appropriate FL solver, our primary objective turns to devise a $\{p_1, \ldots, p_T\}$ that bounds the static regret $\text{Regret}_S(T)$ in Equation 7.

## 3.2 APPROACHING SUB-OPTIMAL PROBABILITY WITH FULL/PARTIAL FEEDBACK

**Full Feedback.** We first investigate the upper bound of $\text{Regret}_S(T)$ in an ideal scenario called full feedback, where the server preserves feedback information of all clients, i.e., $\{\pi_\tau(i)\}_{\tau=1}^{t-1}, \forall i \in [N], t \in [T]$. In practice, the information cannot be obtained exactly, because it requires all clients to compute local updates. Despite that, we can acquire a preliminary solution and extend it into practical settings.

We utilize the classic follow-the-regularized-leader (FTRL) (Shalev-Shwartz et al., 2012; Kalai & Vempala, 2005; Hazan, 2012) framework to design sampling distribution, which is formed at time $t$:

$$p^t := \arg\min_p \left\{ \sum_{\tau=1}^{t-1} \ell_\tau(p) + \sum_{i=1}^{N} \frac{\gamma}{p_i} \right\}, \tag{8}$$

where the regularizer $\gamma$ ensures that the distribution does not change too much and prevents assigning a vanishing probability to any clients. We have the closed-form solution as shown below:

**Lemma 3.3** (Full feedback solution). *Denoting $\pi_{1:t}^2(i) := \sum_{\tau=1}^{t} \pi_\tau^2(i)$ as the cumulative feedback, sorting the regularized feedback denoted by $a_i = \sqrt{\pi_{1:t-1}^2(i) + \gamma}$ in ascending order (i.e., $0 \leq a_1 \leq \cdots \leq a_N$), we utilize Lemma 2.3 to get the solution $p_i^t \propto \sqrt{\pi_{1:t-1}^2(i) + \gamma}$ to Equation 8.*

For $t = 1, \ldots, T$, if applied sampling probability follows Lemma 3.3, we can obtain that $\text{Regret}_S(T)/T \leq \mathcal{O}(1/\sqrt{T})$ as we shown in Theorem C.1. Applying Equation 8 in FedAvg yields a sampling probability sequence that implements sub-optimal profits over time $T$. However, the sub-optimality requiring full feedback is not practical as we only have access to sampled clients in each round. Hence, the partial feedback solution is what we really pursued.

**Partial Feedback.** We extend the full feedback solution to the partial feedback scenario, where the server only has access to the feedback information from the sampled clients. Denoting $\{\pi_t(i)\}_{i \in S^t}$ as partial feedback from sampled points, we construct an additional estimate of the true feedback for all clients denoted by $\tilde{p}$ and let $S^t \sim \tilde{p}^t$, which incurs

$$\tilde{\pi}_t^2(i) := \frac{\pi_t^2(i)}{\tilde{p}_i^t} \cdot \mathbb{I}_{i \in S^t}, \text{and } \mathbb{E}[\tilde{\pi}_t^2(i)|\tilde{p}_i^t] = \pi_t^2(i), \forall i \in [N].$$

Analogously, we define modified cost functions and their unbiased estimates:

$$\tilde{\ell}_t(p) := \sum_{i=1}^N \frac{\tilde{\pi}_t^2(i)}{p_i}, \text{and } \mathbb{E}[\tilde{\ell}_t(p)|\tilde{p}^t, \ell_t] = \ell_t(p).$$

Relying on the additional estimates, the sampling probability $\tilde{p}^t$ can be applied as a partial feedback solution. But, it still depends on $p^t$, which is the distribution from the full feedback scenario in theory. This difference poses a difficulty, where the modified cost functions can be unbounded. To better bound the regrets of estimator $\tilde{p}^t$ in the partial feedback scenario, we mix the original estimator $p^t$ with a static distribution. Let $\theta \in [0, 1]$, we have,

$$\text{Mixing strategy:} \qquad \tilde{p}^t = (1 - \theta)p^t + \theta \frac{K}{N}, \tag{9}$$

where $\tilde{p}^t \geq \theta \frac{K}{N}$, and hence $\tilde{\pi}_t^2(i) \leq \pi_t^2(i) \cdot \frac{N}{\theta K} \leq G^2 \cdot \frac{N}{\theta K}$. The mixing strategy guarantees the least probability that any clients be sampled, thereby encouraging exploration. Besides, the additional estimates transfer our target to bound an expected regret as $\min_p \mathbb{E}[\sum_{t=1}^T \ell_t(\tilde{p}^t) - \sum_{t=1}^T \ell_t(p^t)]$, which denotes the expectation discrepancy between the partial feedback and the full feedback solutions. After analysis detailed in Appendix C.3, we present the expected regret bound of the sampling with mixed probability and the K-Vib sampler outlined in Algorithm 2.

**Theorem 3.4** (Static expected regret with partial feedback). *Under Assumptions 3.1, sampling $S^t \sim \tilde{p}^t$ with $\mathbb{E}[|S^t|] = K$ for all $t = 1, \ldots, T$, and letting $\theta = (\frac{N}{TK})^{1/3}, \gamma = G^2 \frac{N}{K\theta}$ with $T \cdot K \geq N$, we obtain the expected regret,*

$$\frac{1}{N^2}\mathbb{E}\left[\sum_{t=1}^T \ell_t(\tilde{p}^t) - \min_p \sum_{t=1}^T \ell_t(p)\right] \leq \tilde{\mathcal{O}}(N^{\frac{1}{3}}T^{\frac{2}{3}}/K^{\frac{4}{3}}), \tag{10}$$

*where $\tilde{\mathcal{O}}$ hides the logarithmic factors.*

---

**Algorithm 2** K-Vib Sampler

---

**Input:** Num clients $N$, sampling expectation $K$, time $T$, regular factor $\gamma$, and mixing factor $\theta$.
12 Initialize weights $\omega(i) = 0$ for all $i \in [N]$.
13 **for** *time $t \in [T]$* **do**
14     $p_i^t \propto \sqrt{\omega(i) + \gamma}$ // by Lemma 3.3
15     $\tilde{p}_i^t \leftarrow (1 - \theta) \cdot p_i^t + \theta \frac{K}{N}$, for all $i \in [N]$ // mixing
16     Draw $S^t \sim \tilde{p}^t$ and play $S^t$ // independent sampling procedure
17     Receive feedbacks $\pi_t(i)$, and update $\omega(i) \leftarrow \omega(i) + \pi_t^2(i)/\tilde{p}_i^t$ for $i \in S^t$

---

**Summary.** The K-Vib sampler facilitates exploration in the early stages during the federated optimization process while creating a promising sampling distribution with cumulative feedback. Its advantages rely on a tighter formulation of variance obtained via the independent sampling procedure in Equation 4. Utilizing a mixing strategy 9, the K-Vib sampler extends the FTRL to practical partial sampling and feedback scenarios. Finally, it implements a linear speedup $K$ as

shown in Theorem 3.4 comparing with random sampling procedure in a similar manner (Borsos et al., 2018). For computational complexity, the main cost involves sorting the cumulative feedback sequence $\{\omega(i)\}_{i=1}^N$ in Algorithm 2, which will not exceed $\mathcal{O}(N \log N)$ with an adaptive sorting algorithm (Estivill-Castro & Wood, 1992).

## 3.3 Convergence Analysis of Unbiased Sampler in FedAvg

Our ultimate goal is to optimize the benefits of applying the sampling technique in federated optimization. We demonstrate the point by providing unique convergence analyses for Algorithm 1 for arbitrary unbiased client sampling techniques. To be general, we use standard assumptions on the local empirical function $f_i, i \in [N]$ in non-convex federated optimization literature (Reddi et al., 2020; Li et al., 2020; Wang et al., 2020).

**Assumption 3.5** (Smothness). *Each objective $f_i(x)$ for all $i \in [N]$ is L-smooth, inducing that for all $\forall x, y \in \mathbb{R}^d$, it holds $\|\nabla f_i(x) - \nabla f_i(y)\| \leq L\|x - y\|$.*

**Assumption 3.6** (Unbiasedness and Bounded Local Variance). *For each $i \in [N]$ and $x \in \mathbb{R}^d$, we assume the access to an unbiased stochastic gradient $g_i(x)$ of client's true gradient $\nabla F_i(x)$, i.e., $\mathbb{E}_{\xi^t \sim \mathcal{D}_i}[\nabla F_i(x, \xi^t)] = \nabla f_i(x)$. The function $f_i$ have $\sigma_l$-bounded (local) variance i.e., $\mathbb{E}_{\xi_i \sim \mathcal{D}_i}\left[\|\nabla F_i(x, \xi^t) - \nabla f_i(x)\|^2\right] \leq \sigma_l^2$.*

**Assumption 3.7** (Bounded Global Variance). *We assume the weight-averaged global variance is bounded, i.e., $\sum_{i=1}^N \lambda_i \|\nabla f_i(x) - \nabla f(x)\|^2 \leq \sigma_g^2$ for all $x \in \mathbb{R}^d$.*

Moreover, we define important quantities below to clarify the unbiased sampling in our analyses.

**Definition 3.8** (The improvement factor and sampling quality.). For $t = 1, \ldots, T$, under the constraints of communication budget $K$ and local updates statues $\{g_i^t\}_{i \in [N]}$, we define the optimal improvement factor of optimal client sampling $S_*^t$ over uniform sampling $U^t \sim \mathbb{U}$ is defined as:

$$\alpha_*^t = \frac{\mathbb{E}\left[\left\|\sum_{i \in S_*^t} \frac{\lambda_i}{p_i^*} g_i^t - \sum_{i=1}^N \lambda_i g_i^t\right\|^2\right]}{\mathbb{E}\left[\left\|\sum_{i \in U^t} \frac{\lambda_i}{p_i} g_i^t - \sum_{i=1}^N \lambda_i g_i^t\right\|\right]}, \text{with independent sampling } S_*^t \sim p_*^t,$$

and optimal $p_*^t$ is computed following Lemma 2.3. Then, given arbitrary client sampling probability $p^t$, we can define the *quality* of once sampling $S^t$ according to the discrepancy to the optimal:

$$Q(S^t) = \mathbb{E}\left[\left\|\sum_{i \in S^t} \frac{\lambda_i g_i^t}{p_i^t} - \sum_{i \in S_*^t} \frac{\lambda_i g_i^t}{p_i^*}\right\|^2\right]. \tag{11}$$

The factor $\alpha_*^t \in [0, 1]$ denotes the best efficiency that one sampling technique can potentially achieve under the current constraints $K, \{g_i^t\}_{i \in [N]}, \lambda$ in theory. In our analysis, $\alpha_*^t$ denotes the best improvement by applying optimal sampling. Furthermore, we define $Q(S^t)$ to denote the discrepancy between a given sampling and the optimal sampling. We use the term to estimate the quality of one sampling. Thereby, it also connects the regret of the adaptive sampling task. Besides, $Q(S^t) = 0$ indicates the current sampling has achieved the optimal sampling. Now, We are ready to provide the non-convex convergence of Algorithm 1 with a focus on the impacts of unbiased sampling techniques.

**Theorem 3.9** (Non-convex convergence of FedAvg with unbiased sampling). *Under Assumptions 3.5 3.6 3.7, for arbitrary sampling $S^t \sim p^t$ and its unbiased estimates follows Equation 2, and taking upper bound $\mathbb{E}\left[f(x^1) - f(x^{+\infty})\right] \leq M$, $\eta_g = \sqrt{\frac{2M}{T\bar{\beta}}}$, and $\eta_l \leq \min(\frac{1}{R}, \frac{1}{\sqrt{5R}})$, we have the convergence of Algorithm 1,*

$$\min_{t \in [T]} \mathbb{E}\|\nabla f(x^t)\|^2 \leq \sqrt{\frac{8M\bar{\beta}}{T\hat{\rho}^2}} + \frac{\frac{1}{T}\sum_{t=1}^T Q(S^t) + \epsilon}{\hat{\rho}}, \tag{12}$$

*where we define*

$$\rho^t := \left(1 - 4(1 - \eta_l R)^2 - 12\eta_l^2 L^2 - 6\eta_l^2 \gamma_*^t W\right), \beta^t := 2L\gamma_*^t W(\sigma_l^2 + 3\sigma_g^2),$$

$$\epsilon := 4\left((1 - \eta_l R)^2 + 3\eta_l^2 L^2\right)\sigma_g^2 + 2\eta_l^2 \left(2L^2 + 1\right)\sigma_l^2,$$

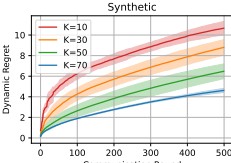 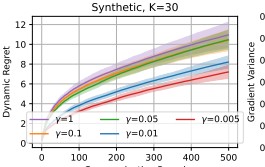 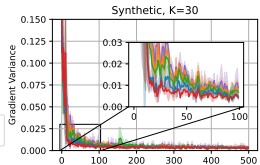

Figure 1: Sub-optimal in variance reduction

Figure 2: Regret improvement with $K$

Figure 3: Sensitivity to regularization $\gamma$

$\gamma_*^t := \frac{\alpha_*^t(N-K)+K}{K} \in [1, \frac{N}{K}]$, $\hat{\rho} := \min\{\rho^t\}_{t=1}^T$, and $\bar{\beta} := \frac{1}{T}\sum_{t=1}^T \beta^t$. Notably, the $\gamma_*^t$ denotes the benefits of utilizing optimal sampling respecting the communication budget $K$. And, $\rho^t, \beta^t$ and $\epsilon$ absorb the learning rate techniques. It allows us to decouple the impacts of sampling quality $Q(S^t)$.

**Remark.** The Equation 12 connects the expected regret in adaptive sampling with the convergence rate of federated optimization to show the impacts of adaptive sampling techniques. For example, we can combine $\frac{1}{T}\sum_{t=1}^T Q(S^t)$ with Definition 11 and Theorem 3.4 to know that $\frac{1}{T}\sum_{t=1}^T Q(S^t) \leq \tilde{\mathcal{O}}(N^{\frac{1}{3}}/T^{\frac{1}{3}}K^{\frac{4}{3}})$. Comparing with previous bound $\tilde{\mathcal{O}}(N^{\frac{1}{3}}T^{\frac{2}{3}})$ (Borsos et al., 2018) and $\mathcal{O}(N^{\frac{1}{3}}T^{\frac{2}{3}})$ (El Hanchi & Stephens, 2020), applying K-Vib sampler in FL achieves faster convergence accordingly. Technically, the theory also provides a fair comparison for unbiased samplers within the framework of Algorithm 1. Moreover, the Equation 12 matches the best-known sub-linear convergence rate $\mathcal{O}(1/\sqrt{T})$ in the non-convex federated optimization (Li et al., 2019; Reddi et al., 2020; Li et al., 2020), and hence verifies the rationality of our Assumption 3.1.

## 4 EXPERIMENTS

We evaluate the theoretical results via experiments on Synthetic datasets, where the data are generated from Gaussian distributions (Li et al., 2020) and the model is logistic regression $y = \arg\max(W^T X + b)$. We generate $N = 100$ clients of each has a synthetic dataset, where the size of each dataset follows the power law. We also evaluate the proposed sampler on the standard federated learning tasks Federated EMNIST (FEMNIST) from LEAF (Caldas et al., 2018). To better illustrate our theoretical improvement, we use the FEMNIST tasks involving three degrees of unbalanced level (Chen et al., 2020), including FEMNIST v1 (10% clients hold 82% training images), FEMNIST v2 (20% client hold 90% training images) and FEMNIST v3 (50% client hold 98% training images). We use the same CNN model as the one used in (McMahan et al., 2017). The data distributions across clients are shown in Appendix, Figure 6.

**Baselines.** We demonstrate our improvement by comparison with the uniform sampling and other "free" adaptive samplers including Multi-armed Bandit Sampler (Mabs) (Salehi et al., 2017), Variance Reducer Bandit (Vrb) (Borsos et al., 2018) and Avare (El Hanchi & Stephens, 2020). As our focus falls on sampling, we run $T = 500$ round for all tasks and use vanilla SGD optimizers with constant step size for both clients and the server, with $\eta_g = 1$. To ensure a fair comparison, we set the hyperparameters of all samplers to the optimal values prescribed in their respective original papers, and the choice of hyperparameters is detailed in the Appendix. We run experiments with the same random seed and vary the seeds across different runs. We present the mean performance with the standard deviation calculated over five independent runs. The experiment implementations are supported by *FedLab* framework (Zeng et al., 2023).

**Theory Evaluation.** We evaluate our theory on the Synthetic dataset task by setting local learning rate $\eta_l = 0.02$, local epoch 1, and batch size 64. We utilize three metrics: (a) dynamic regret as defined in Equation 6, (b) gradient variance in Equation 2, and (c) loss value on the test dataset. Our theoretical evaluation is to demonstrate the following three points in our theory.

**1) Approaching sub-optimal estimation.** We use gradient variance as a metric to demonstrate our theory of implementing a sub-optimal estimation. The results are presented in Figure 1. The discrepancies between the optimal curve and the full-feedback curve demonstrate the vanishing gaps given in Theorem 3.2. Besides, the vanishing differences between the full-feedback curve and the partial-feedback curve prove that the K-Vib sampler implements a promising performance by approaching the full-feedback results, as we proved in Theorem 3.4.

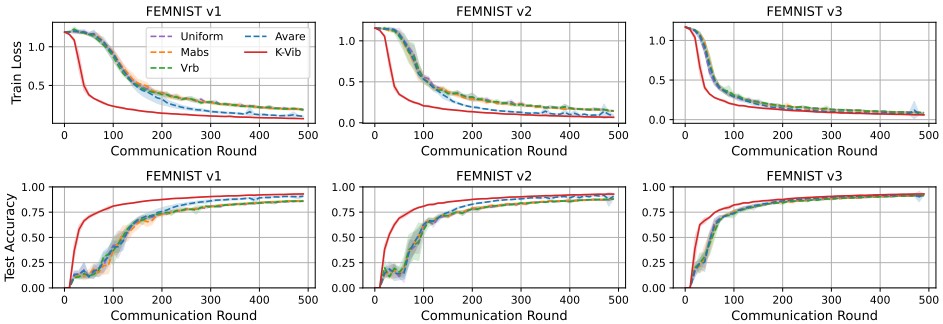

Figure 4: Evaluation of samplers with dynamic regret (left), variance (middle), and test loss (right). K-Vib outperforms baselines by establishing a lower dynamic regret. This process minimizes the gradient variance and hence enables a faster convergence.

Figure 5: Training loss and test accuracy of FedAvg with different unbiased samplers. We observe that the K-Vib converges faster at early rounds. This is because the lower variance in estimates induces faster convergence in Theorem 3.9 and the variance of K-Vib is lower compared with baselines at the beginning as shown in Lemma 2.2. Meanwhile, the K-Vib sampler further enlarges the convergence benefits during the training process and hence maintains the fast convergence speed. Horizontally comparing the results, we observe that the discrepancies between K-Vib and baselines match the degrees of variance across datasets. The variance slows the convergence of vanilla FedAvg but is mitigated by the K-Vib sampler.

**2) Speed up $K$ and regularization $\gamma$.** We present Figure 2 to prove the linear speed up in Theorem 3.4. In detail, with the increase of budget $K$, the performance of the K-Vib sampler with regret metric is reduced significantly. Due to page limitation, we provide further illustration examples of other baselines in the same metric in the Appendix F. The results demonstrate our unique improvements in theory. Besides, Figure 3 reveals the effects of regularization $\gamma$ in Algorithm 2. The variance reduction curve indicates that the K-Vib sampler is not sensitive to $\gamma$ in the task.

**3) Variance reduction comparison.** We present the results with $K = 10$ in Figure 4 to showcase our improvement with baseline samplers. The K-Vib outperforming baselines on online metrics prove our theoretical improvement. Moreover, the variance of global estimates is significantly reduced. Hence, the K-Vib achieves faster convergence shown in the test loss curve. We present additional results respecting different $K$ values in Appendix F, where we observe the same phenomenon in Figure 4.

**Federated Optimization Evaluation.** We present the evaluation results of the FEMNIST tasks with communication round $T = 500$, batch size 20, local epochs 3, $\eta_l = 0.01$, and $K = 111, 62, 23$ as 5% of total clients. We report the convergence performance on FEMNIST tasks in Figure 5.

## 5 CONCLUSION AND FUTURE WORK

In this paper, we extended the line of unbiased sampling techniques in stochastic optimization and explored its application on unbiased client sampling in federated optimization. Based on the observation of the sampling procedure, we present an efficient K-Vib sampler that achieves a linear speed up in online convex optimization metric and the best performance in classic FL tasks comparing baselines. The mixing strategy can be improved by designing an optimization task on the server to find the best additional estimation, which we will explore in future work. Besides, we will further study the ability of the proposed sampler in cooperation with advanced federated learning algorithms, such as adaptive optimization and learning rate techniques.

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

# Appendices

# A    RELATED WORK

Our paper contributes to the literature on the importance sampling in stochastic optimization, online convex optimization, and client sampling in FL.

**Importance Sampling.** Importance sampling is a non-uniform sampling technique widely used in stochastic optimization (Katharopoulos & Fleuret, 2018) and coordinate descent (Richtárik & Takáč, 2016a). Zhao & Zhang (2015); Needell et al. (2014) connects the variance of the gradient estimates and the optimal sampling distribution is proportional to the per-sample gradient norm. The insights of sampling and optimization quality can be transferred into federated client sampling, as we summarised below.

**Online Variance Reduction.** Our paper addresses the topic of online convex optimization for reducing variance. Variance reduction techniques are frequently used in conjunction with stochastic optimization algorithms (Defazio et al., 2014; Johnson & Zhang, 2013) to enhance optimization performance. These same variance reduction techniques have also been proposed to quicken federated optimization (Dinh et al., 2020; Malinovsky et al., 2022). On the other hand, online learning (Shalev-Shwartz et al., 2012) typically employs an exploration-exploitation paradigm to develop decision-making strategies that maximize profits. Although some studies have considered client sampling as a multi-armed bandit problem, they have only provided limited theoretical results (Kim et al., 2020; Cho et al., 2020a; Yang et al., 2021). In an intriguing combination, certain studies (Salehi et al., 2017; Borsos et al., 2018; 2019) have formulated data sampling in stochastic optimization as an online learning problem. These methods were also applied to client sampling in FL by treating each client as a data sample in their original problem (Zhao et al., 2021; El Hanchi & Stephens, 2020).

**Client Sampling in FL.** Client sampling methods in FL fall under two categories: biased and unbiased methods. Unbiased sampling methods ensure objective consistency in FL by yielding the same expected value of results as global aggregation with the participation of all clients. In contrast, biased sampling methods converge to arbitrary sub-optimal outcomes based on the specific sampling strategies utilized. Additional discussion about biased and unbiased sampling methods is provided in Appendix E.2. Recent research has focused on exploring various client sampling strategies for both biased and unbiased methods. For instance, biased sampling methods involve sampling clients with probabilities proportional to their local dataset size (McMahan et al., 2017), selecting clients with a large update norm with higher probability (Chen et al., 2020), choosing clients with higher losses (Cho et al., 2020b), and building a submodular maximization to approximate the full gradients (Balakrishnan et al., 2022). Meanwhile, several studies (Chen et al., 2020; Cho et al., 2020b) have proposed theoretically optimal sampling methods for FL utilizing the unbiased sampling framework, which requires all clients to upload local information before conducting sampling action. Moreover, cluster-based sampling (Fraboni et al., 2021; Xu et al., 2021; Shen et al., 2022) relies on additional clustering operations where the knowledge of utilizing client clustering can be transferred into other client sampling techniques.

# B    IMPORTANT CONCEPTS AND LEMMAS

## B.1    ARBITRARY SAMPLING

In this section, we summarize the arbitrary sampling techniques and present key lemmas used in this paper. Arbitrary sampling paradigm (Chambolle et al., 2018; Richtárik & Takáč, 2016a; Qu & Richtárik, 2016) is used either for generating mini-batches of samples in stochastic algorithms or for coordinate descent optimization.

In detail, let $S$ denote a sampling, which is a random set-valued mapping with values in $2^{[N]}$, where $[N] := \{1, 2, \ldots, N\}$. An arbitrary sampling $S$ is generated by assigning probabilities to all $2^N$ subsets of $[N]$, which associates a *probability matrix* $\mathbf{P} \in \mathbb{R}^{N \times N}$ defined by

$$\mathbf{P}_{ij} := \text{Prob}(\{i, j\} \subseteq S).$$

Thus, the *probability vector* $p = (p_1, \ldots, p_N) \in \mathbb{R}^N$ is composed of the diagonal entries of $\mathbf{P}$, and $p_i := \text{Prob}(i \in S)$. Furthermore, we say that $S$ is *proper* if $p_i > 0$ for all $i$. Thus, it incurs that

$$K := \mathbb{E}[|S|] = \text{Trace}(\mathbf{P}) = \sum_{i=1}^{N} p_i.$$

The definition of sampling can be naively transferred to the context of federated client sampling. We refer to $K$ as the expected number of sampled clients per round in FL. The following lemma plays a key role in our problem formulation and analysis.

**Lemma B.1** (Generalization of Lemma 1 Horváth & Richtárik (2019)). *Let $a_1, a_2, \ldots, a_N$ be vectors in $\mathbb{R}^d$ and let $\bar{a} = \sum_{i=1}^{N} \lambda_i a_i$ be their weighted average. Let $S$ be a proper sampling. Assume that there is $v \in \mathbb{R}^N$ such that*

$$\mathbf{P} - pp^T \preceq \boldsymbol{Diag}(p_1 v_1, p_2 v_2, \ldots, p_N v_N). \tag{13}$$

*Then, we have*

$$\mathbb{E}_{S \sim p}\left[\left\|\sum_{i \in S} \frac{\lambda_i a_i}{p_i} - \bar{a}\right\|^2\right] \leq \sum_{i=1}^{N} \lambda_i^2 \frac{v_i}{p_i} \|a_i\|^2, \tag{14}$$

*where the expectation is taken over sampling $S$. Whenever Equation equation 13 holds, is must be the case that*

$$v_i \geq 1 - p_i.$$

*Moreover, The random sampling admits $v_i = \frac{N-K}{N-1}$. The independent sampling admits $v_i = 1 - p_i$ and makes Equation equation 14 hold as equality.*

*Proof.* Let $\mathbb{I}_{i \in S} = 1$ if $i \in S$ and $\mathbb{I}_{i \in S} = 0$ otherwise. Similarly, let $\mathbb{I}_{i,j \in S} = 1$ if $i \in S$ and $\mathbb{I}_{i,j \in S} = 0$ otherwise. Note that $\mathbb{E}[\mathbb{I}_{i \in S}] = p_i$ and $\mathbb{E}[\mathbb{I}_{i,j \in S}] = \mathbf{P}_{ij}$. Then, we compute the mean of estimates $\tilde{a} := \sum_{i \in S} \frac{\lambda_i a_i}{p_i}$:

$$\mathbb{E}[\tilde{a}] = \mathbb{E}[\sum_{i \in S} \frac{\lambda_i a_i}{p_i}] = \mathbb{E}[\sum_{i=1}^{N} \frac{\lambda_i a_i}{p_i} \mathbb{I}_{i \in S}] = \sum_{i=1}^{N} \frac{\lambda_i a_i}{p_i} \mathbb{E}[\mathbb{I}_{i \in S}] = \sum_{i=1}^{N} \lambda_i a_i = \bar{a}.$$

Let $\mathbf{A} = [\zeta_1, \ldots, \zeta_N] \in \mathbb{R}^{d \times N}$, where $\zeta_i = \frac{\lambda_i a_i}{p_i}$, and let $e$ be the vector of all ones in $\mathbb{R}^N$. We now write the variance of $\tilde{a}$ in a form that will be convenient to establish a bound:

$$\begin{aligned}
\mathbb{E}[\|\tilde{a} - \mathbb{E}[\tilde{a}]\|^2] &= \mathbb{E}[\|\tilde{a}\|^2] - \|\mathbb{E}[\tilde{a}]\|^2 \\
&= \mathbb{E}[\|\sum_{i \in S} \frac{\lambda a_i}{p_i}\|^2] - \|\bar{a}\|^2 \\
&= \mathbb{E}\left[\sum_{i,j} \frac{\lambda_i a_i^\top}{p_i} \frac{\lambda_j a_j}{p_j} \mathbb{I}_{i,j \in S}\right] - \|\bar{a}\|^2 \\
&= \sum_{i,j} p_{ij} \frac{\lambda_i a_i^\top}{p_i} \frac{\lambda_j a_j}{p_j} - \sum_{i,j} \lambda_i \lambda_j a_i^\top a_j \\
&= \sum_{i,j} (p_{ij} - p_i p_j) \zeta_i^\top \zeta_j \\
&= e^\top \left((\mathbf{P} - pp^\top) \circ \mathbf{A}^\top \mathbf{A}\right) e.
\end{aligned} \tag{15}$$

Since by assumption we have $\mathbf{P} - pp^\top \preceq \mathbf{Diag}(p \circ v)$, we can further bound

$$e^\top \left((\mathbf{P} - pp^\top) \circ \mathbf{A}^\top \mathbf{A}\right) e \leq e^\top \left(\mathbf{Diag}(p \circ v) \circ \mathbf{A}^\top \mathbf{A}\right) e = \sum_{i=1}^{n} p_i v_i \|\zeta_i\|^2. \tag{16}$$

To obtain equation 14, it remains to combine equation 16 with equation 15. Since $\mathbf{P} - pp^\top$ is positive semi-definite (Richtárik & Takáč, 2016b), we can bound $\mathbf{P} - pp^\top \preceq N\mathbf{Diag}(\mathbf{P} - pp^\top) = \mathbf{Diag}(p \circ v)$, where $v_i = N(1 - p_i)$.

Overall, arbitrary sampling that associates with a probability matrix $\mathbf{P}$ will determine the value of $v$. As a result, we summarize independent sampling and random sampling as follows,

- Consider now the independent sampling,

$$\mathbf{P} - pp^\top = \begin{bmatrix} p_1\,(1-p_1) & 0 & \cdots & 0 \\ 0 & p_2\,(1-p_2) & \cdots & 0 \\ \vdots & \vdots & \ddots & \vdots \\ 0 & 0 & \cdots & p_n\,(1-p_n) \end{bmatrix} = \mathbf{Diag}\,(p_1 v_1, \ldots, p_n v_n),$$

where $v_i = 1 - p_i$. Therefore, independent sampling always minimizes equation 14, making it hold as equality.

- Consider the random sampling,

$$\mathbf{P} - pp^\top = \begin{bmatrix} \frac{K}{N} - \frac{K^2}{N^2} & \frac{K(K-1)}{N(N-1)} & \cdots & \frac{K(K-1)}{N(N-1)} \\ \frac{K(K-1)}{N(N-1)} & \frac{K}{N} & \cdots & \frac{K(K-1)}{N(N-1)} \\ \vdots & \vdots & \ddots & \vdots \\ \frac{K(K-1)}{N(N-1)} & \frac{K(K-1)}{N(N-1)} & \cdots & \frac{K}{N} \end{bmatrix}.$$

As shown in (Horváth & Richtárik, 2019), the standard random sampling admits $v_i = \frac{N-K}{N-1}$ for equation 14.

$\square$

***Conclusion.*** Given probabilities $p$ that defines all samplings $S$ satisfying $p_i = \mathrm{Prob}(i \in S)$, it turns out that the independent sampling (i.e., $\mathbf{P}_{ij} = \mathrm{Prob}(i, j \in S) = \mathrm{Prob}(i \in S)\mathrm{Prob}(j \in S) = p_i p_j$) minimizes the upper bound in Equation equation 14. Therefore, depending on the sampling distribution and method, we can rewrite the Equation equation 14 as follow:

$$\mathbb{V}(a, p, \lambda) = \mathbb{E}_{S \sim p}[\|\sum_{i \in S} \frac{\lambda_i a_i}{a_i} - \bar{a}\|^2] = \underbrace{\sum_{i=1}^{N}(1 - p_i)\frac{\lambda_i^2\|a_i\|^2}{p_i}}_{\text{Independent Sampling}} \leq \underbrace{\frac{N-K}{N-1}\sum_{i=1}^{N}\frac{\lambda_i^2\|a_i\|^2}{p_i}}_{\text{Random Sampling}}. \quad (17)$$

## B.2 AUXILIARY LEMMAS

Here are some common inequalities used in our analysis.

**Lemma B.2.** *For an arbitrary set of $n$ vectors $\{a_i\}_{i=1}^n, a_i \in \mathbb{R}^d$,*

$$\left\|\sum_{i=1}^{n} \mathbf{a}_i\right\|^2 \leq n \sum_{i=1}^{n} \|\mathbf{a}_i\|^2. \quad (18)$$

**Lemma B.3.** *For random variables $z_1, \ldots, z_r$, we have*

$$\mathbb{E}\left[\|z_1 + \ldots + z_r\|^2\right] \leq r\mathbb{E}\left[\|z_1\|^2 + \ldots + \|z_r\|^2\right]. \quad (19)$$

**Lemma B.4.** *For independent, mean 0 random variables $z_1, \ldots, z_r$, we have*

$$\mathbb{E}\left[\|z_1 + \ldots + z_r\|^2\right] = \mathbb{E}\left[\|z_1\|^2 + \ldots + \|z_r\|^2\right]. \quad (20)$$

## B.3 USEFUL LEMMAS AND COROLLARIES

In this section, we present some useful lemmas and their proofs for our theoretical analysis. We first offer proof details for Lemma 2.3 with a general constraint. Then, we provide several Corollaries B.7 B.8 B.9 for our analysis in the next section.

**Lemma B.5.** *Let $0 < a_1 \le a_2 \le \cdots \le a_N$ and $0 < K \le N$. We consider the following optimization objective with a restricted probability space $\Delta = \{p \in \mathbb{R}^N | p_{min} \le p_i \le 1, \sum_{i=1}^N p_i = K, \forall i \in [N]\}$ where $p_{min} \le K/N$,*

$$\begin{aligned}
minimize_{p \in \Delta} \; \Omega(p) &= \sum_{i=1}^N \frac{a_i^2}{p_i} \\
subject\ to \; \sum_{i=1}^N p_i &= K, \\
p_{min} \le p_i \le 1, \; &i = 1, 2, \dots, N.
\end{aligned} \tag{21}$$

*Proof.* We formulate the Lagrangian:

$$\mathcal{L}(p, y, \alpha_1, \dots, \alpha_N, \beta_1, \dots, \beta_N) = \sum_{i=1}^N \frac{a_i^2}{p_i} + y \cdot \left( \sum_{i=1}^N p_i - K \right) + \sum_{i=1}^N \alpha_i (p_{\min} - p_i) + \sum_{i=1}^N \beta_i (p_i - 1). \tag{22}$$

The constraints are linear and KKT conditions hold. Hence, we have,

$$p_i = \sqrt{\frac{a_i^2}{y - \alpha_i + \beta_i}} = \begin{cases} 1, & \text{if } \sqrt{y} \le a_i. \\ \sqrt{\frac{a_i^2}{y}}, & \text{if } \sqrt{y} \cdot p_{\min} < a_i < \sqrt{y}, \\ p_{\min}, & \text{if } a_i \le \sqrt{y} \cdot p_{\min}. \end{cases} \tag{23}$$

Then, we analyze the value of $y$. Letting $l_1 = \left| \{i | a_i \le \sqrt{y} \cdot p_{\min}\} \right|, l_2 = l1 + |\{\sqrt{y} \cdot p_{\min} < a_i < \sqrt{y}\}|$, $N - l_2 = \left| \{i | \sqrt{y} \le a_i\} \right|$, and using $\sum_{i=1}^N p_i = K$ implies,

$$\sum_{i=1}^N p_i = \sum_{i \le l_1} p_i + \sum_{l_1 < i < l_2} p_i + \sum_{i \ge l_2} p_i = l_1 \cdot p_{\min} + \sum_{l_1 < i < l_2} \sqrt{\frac{a_i^2}{y}} + N - l_2 = K.$$

Arrange the formula, we get

$$\sqrt{y} = \frac{\sum_{l_1 < i < l_2} a_i}{K - N + l_2 - l_1 \cdot p_{\min}}. \tag{24}$$

Moreover, we can plug the results into the objective to get the optimal result:

$$\begin{aligned}
\sum_{i=1}^N \frac{a_i^2}{p_i} &= \sum_{i \le l_1} \frac{a_i^2}{p_i} + \sum_{l_1 < i < l_2} \frac{a_i^2}{p_i} + \sum_{i \ge N - l_2} \frac{a_i^2}{p_i} \\
&= \frac{\sum_{i \le l_1} a_i^2}{p_{\min}} + \sqrt{y} \left( \sum_{l_1 < i < l_2} a_i \right) + \sum_{i \ge N - l_2} a_i^2 \\
&= \frac{\sum_{i \le l_1} a_i^2}{p_{\min}} + \frac{(\sum_{l_1 < i < l_2} a_i)^2}{K - N + (l_2 - l_1 \cdot p_{\min})} + \sum_{i \ge N - l_2} a_i^2,
\end{aligned} \tag{25}$$

where the $1 \le l_1 \le l_2 \le N$ satisfies that $\forall i \in (l_1, l_2)$,

$$p_{\min} \cdot \frac{\sum_{l_1 < i < l_2} a_i}{K - N + l_2 - l_1 \cdot p_{\min}} < a_i < \frac{\sum_{l_1 < i < l_2} a_i}{K - N + l_2 - l_1 \cdot p_{\min}}.$$

In short, we note that if let $p_{\min} = 0, l_1 = 0$, the Lemma B.6/Lemma 2.3 is proved as a special case of Equation 25. Besides, we provide further Corollary B.8 and B.9 as preliminaries for further analysis.

**Lemma B.6.** *Let $0 < a_1 \leq a_2 \leq \cdots \leq a_N$ and $0 < K \leq N$. We consider the following optimization objective,*

$$minimize_{p \in \mathbb{R}^N} \; \Omega(p) = \sum_{i=1}^{N} \frac{a_i^2}{p_i}$$

$$subject \; to \; \sum_{i=1}^{N} p_i = K, \tag{26}$$

$$0 \leq p_i \leq 1, \; i = 1, 2, \ldots, N.$$

*Then, we have*

$$p_i^* = \begin{cases} (K + l - N)\frac{a_i}{\sum_{j=1}^{l} a_j}, & \text{if } i \leq l, \\ 1, & \text{if } i > l, \end{cases} \tag{27}$$

*where $l$ is the largest integer for which $0 < K + l - N \leq \frac{\sum_{i=1}^{l} a_i}{a_l}$.*

Connecting with the aforementioned assumptions, we provide an additional corollary below for further analysis.

**Corollary B.7.** *We note that $K \cdot a_N \leq \sum_{i=1}^{N} a_i$ can be always satisfied, if functions $a_i, i \in [N]$ is L-smooth. Thus, letting $l = N$, we have*

$$\arg \min \Omega(p^*) = \frac{(\sum_{i=1}^{N} a_i)^2}{K}.$$

**Corollary B.8.** *Following Corollary B.7 and the assumption that functions $a_i, i \in [N]$ is L-smooth, thus $l_2 = N$, and we can know that $l_1$ is the largest integer that satisfies $0 < (K - l_1 \cdot p_{min})\frac{a_{l_1}}{\sum_{i=l_1}^{N} a_i} < p_{min}$. The optimal value of the objective is,*

$$\arg \min \Omega(p^*) = \frac{\sum_{i \leq l_1} a_i^2}{p_{min}} + \frac{(\sum_{i=l_1+1}^{N} a_i)^2}{K - l_1 \cdot p_{min}}.$$

*We note that $p_{min} = 0$ incurs $l_1 = 1$ will make this corollary return to Corollary B.7.*

**Corollary B.9.** *Letting $l_2 = N$ and following the assumption in Corollary B.8, we further bound the value of the objective in Equation equation 21,*

$$\begin{aligned}
\sum_{i=1}^{N} \frac{a_i^2}{p_i} &= \frac{\sum_{i \leq l_1} a_i^2}{p_{min}} + \sqrt{y}(\sum_{l_1 < i \leq N} a_i) && \triangleright \textit{Eq. equation 25, def. in line 2} \\
&= \frac{\sum_{i \leq l_1} a_i^2}{p_{min}} + y(K - l_1 p_{min}) && \triangleright \textit{Eq. 24, replacing } \sum_{l_1 < i \leq N} a_i \\
&\leq l_1 y p_{min} + y(K - l_1 p_{min}) && \triangleright \textit{Eq. 23, } a_i \leq \sqrt{y} \cdot p_{min} \\
&= \frac{(\sum_{i=l_1}^{N} a_i)^2}{(K - l_1 p_{min})^2} \cdot K \leq \frac{K(\sum_{i=l_1}^{N} a_i)^2}{(K - N p_{min})^2} \\
&\leq \frac{K(\sum_{i=1}^{N} a_i)^2}{(K - N p_{min})^2}.
\end{aligned}$$

$\square$

# C DETAIL PROOFS OF ONLINE CONVEX OPTIMIZATION FOR GRADIENT VARIANCE REDUCTION

## C.1 VANISING HINDSIGHT GAP: PROOF OF LEMMA 3.2

*Proof.* We first arrange the term (B) in Equation equation 7 as follows,

$$\frac{1}{N^2}\left(\min_p \sum_{t=1}^{T} \ell_t(p) - \sum_{t=1}^{T} \min_p \ell_t(p)\right) = \min_p \frac{1}{N^2} \sum_{t=1}^{T}\sum_{i=1}^{N} \frac{\pi_t^2(i)}{p_i} - \sum_{t=1}^{T} \min_p \frac{1}{N^2}\sum_{i=1}^{N} \frac{\pi_t^2(i)}{p_i}. \quad (28)$$

Here, we recall our mild Assumption 3.1,

$$\pi_*(i) := \lim_{t \to \infty} \pi_t(i),\ \Pi_* := \frac{1}{N}\sum_{i=1}^{N} \pi_*(i),\ \forall i \in [N].$$

Then, denoting $V_T(i) := \sum_{t=1}^{T}(\pi_t(i) - \pi_*(i))^2$, we bound the cumulative variance over time $T$ per client $i \in [N]$,

$$\begin{aligned}
\pi_{1:T}^2(i) &= \sum_{t=1}^{T}(\pi_*(i) + (\pi_t(i) - \pi_*(i)))^2 \\
&\leq T \cdot \pi_*^2(i) + 2\pi_*(i)\sum_{t=1}^{T}|\pi_t(i) - \pi_*(i)| + \sum_{t=1}^{T}(\pi_t(i) - \pi_*(i))^2 \\
&\leq T \cdot \pi_*^2(i) + 2\pi_*(i)\sqrt{T \cdot V_T(i)} + V_T(i) \\
&= T\left(\pi_*(i) + \sqrt{V_T(i)/T}\right)^2.
\end{aligned} \quad (29)$$

Using the Lemma B.6 and non-negativity of feedback we have,

$$\min_p \sum_{i=1}^{N} \frac{\pi_t^2(i)}{p_i} = \frac{(\sum_{i=1}^{N} \pi_t(i))^2}{K}. \quad (30)$$

We obtain the upper bound of the first term in Equation equation 28,

$$\begin{aligned}
\min_p \frac{1}{N^2}\sum_{t=1}^{T}\sum_{i=1}^{N}\frac{\pi_t^2(i)}{p_i} &= \min_p \frac{1}{N^2}\sum_{i=1}^{N}\frac{\pi_{1:T}^2(i)}{p_i} = \frac{\left(\sum_{i=1}^{N}\sqrt{\pi_{1:T}^2(i)}\right)^2}{N^2 K} \\
&\leq \frac{T}{K}\left(\frac{1}{N}\sum_{i=1}^{N}\pi_*(i) + \frac{1}{N}\sum_{i=1}^{N}\sqrt{\frac{V_T(i)}{T}}\right)^2 \\
&= \left(T\Pi_*^2 + 2\sqrt{T}\Pi_*\frac{1}{N}\sum_{i=1}^{N}\sqrt{V_T(i)} + \left(\frac{1}{N}\sum_{i=1}^{N}\sqrt{V_t(i)}\right)^2\right)/K,
\end{aligned} \quad (31)$$

where we use Lemma B.6 in the second line, and Equation equation 29 in the third line.

Then, we bound the second term in Equation equation 28:

$$\begin{aligned}
N^2 \cdot \Pi_*^2 = \sum_{i=1}^{N}\pi_*^2(i) &\leq \left(\frac{1}{T}\sum_{t=1}^{T}\sum_{i=1}^{N}\pi_t(i)\right)^2 \leq \frac{1}{T}\sum_{t=1}^{T}(\sum_{i=1}^{N}\pi_t(i))^2 \\
&= \frac{K}{T}\sum_{t=1}^{T}\min_p \sum_{i=1}^{N}\frac{\pi_t^2(i)}{p_i},
\end{aligned} \quad (32)$$

where the first line uses the average assumption, the third line uses Jensen's inequality, and the last line uses Equation equation 30.

Overall, we combine the results in Equation equation 31 and equation 32, and conclude the proof:

$$\frac{1}{N^2}\left[\min_p \sum_{t=1}^T \ell_t(p) - \sum_{t=1}^T \min_p \ell_t(p)\right] \le 2\sqrt{T}\Pi_* \frac{1}{NK}\sum_{i=1}^N \sqrt{V_T(i)} + \left(\frac{1}{NK}\sum_{i=1}^N \sqrt{V_T(i)}\right)^2. \quad (33)$$

$\square$

## C.2 REGRET OF FULL FEEDBACK

**Theorem C.1** (Static regret with full feedback). *Under Assumptions 3.1, sampling a batch of clients with an expected size of $K$, and setting $\gamma = G^2$, the FTRL scheme in Equation 8 yields the following regret,*

$$\frac{1}{N^2}\left(\sum_{t=1}^T \ell_t(p^t) - \min_p \sum_{t=1}^T \ell_t(p)\right) \le \frac{27G}{NK}\sum_{i=1}^N \sqrt{\pi_{1:T}^2(i)} + \frac{22G^2}{K}, \quad (34)$$

*where we note the cumulative feedback $\sqrt{\pi_{1:T}^2(i)} \le \mathcal{O}(\sqrt{T})$ following Assumption 3.1.*

*Proof.* We considering a restricted probability space $\Delta = \{p \in \mathbb{R}^N | p_i \ge p_{\min}, \sum_{i=1}^N p_i = K, \forall i \in [N]\}$ where $p_{\min} \le K/N$. Then, we decompose the regret,

$$N^2 \cdot \text{Regret}_{\text{FTRL}}(T) = \underbrace{\sum_{t=1}^T \ell_t(p^t) - \min_{p \in \Delta}\sum_{t=1}^T \ell_t(p)}_{(A)} + \underbrace{\min_{p \in \Delta}\sum_{t=1}^T \ell_t(p) - \min_p \sum_{t=1}^T \ell_t(p)}_{(B)}. \quad (35)$$

We separately bound the above terms in this section. The bound of (A) is related to the stability of the online decision sequence by playing FTRL, which is given in Lemma C.2. Term (B) is bounded by the minimal results of directing calculation.

**Bounding (A).** Without loss of generality, we introduce the stability of the online decision sequence from FTRL to variance function $\ell$ as shown in the following lemma(Kalai & Vempala, 2005) (proof can also be found in (Hazan, 2012; Shalev-Shwartz et al., 2012)).

**Lemma C.2.** *Let $\mathcal{K}$ be a convex set and $\mathcal{R}: \mathcal{K} \mapsto \mathbb{R}$ be a regularizer. Given a sequence of functions $\{\ell_t\}_{t\in[T]}$ defined over $\mathcal{K}$, then setting $p^t = \arg\min_{p\in\mathbb{R}^N}\sum_{\tau=1}^{t-1}\ell_\tau(p) + \mathcal{R}(p)$ ensures,*

$$\sum_{t=1}^T \ell_t(p^t) - \sum_{t=1}^T \ell_t(p) \le \sum_{t=1}^T (\ell_t(p^t) - \ell_t(p^{t+1})) + (\mathcal{R}(p) - \mathcal{R}(p^1)), \forall p \in \mathcal{K}.$$

We note that $\mathcal{R}(p) = \sum_{i=1}^N \gamma/p_i$ in our work. Furthermore, $\mathcal{R}(p)$ is non-negative and bounded by $N\gamma/p_{\min}$ with $p \in \Delta$. Thus, the above lemma incurs,

$$\sum_{t=1}^T \ell_t(p^t) - \sum_{t=1}^T \ell_t(p) \le \underbrace{\sum_{t=1}^T (\ell_t(p^t) - \ell_t(p^{t+1}))}_{\text{Bounded Below}} + \frac{N\gamma}{p_{\min}}. \quad (36)$$

To simply the following proof, we assume that $0 < f_1(t) \le f_2(t) \le \cdots \le f_N(t), t \in [T]$ to satisfies Lemma B.6 without the loss of generality.

We recall that the closed form solution for the probability $p_i^t$ of the $i$-th client at the time $t$ in Lemma B.6,

$$p_i^t = \frac{K\sqrt{f_{1:t-1}^2(i) + \gamma}}{c_t},$$

where $c_t = \sum_{i=1}^{N} \sqrt{f_{1:t-1}^2(i) + \gamma}$ is the normalization factor. Noting that $\{c_t\}_{t \in [T]}$ is a non-decreasing sequence. Then, we further bound the first term in the above inequality,

$$
\begin{aligned}
\sum_{t=1}^{T} (\ell_t(p^t) - \ell_t(p^{t+1})) &= \sum_{t=1}^{T} \sum_{i=1}^{N} \pi_t^2(i) \cdot \left( \frac{1}{p_i^t} - \frac{1}{p_i^{t+1}} \right) \\
&= \sum_{t=1}^{T} \sum_{i=1}^{N} \frac{\pi_t^2(i)}{K} \cdot \left( \frac{c_t}{\sqrt{f_{1:t-1}^2(i) + \gamma}} - \frac{c_{t+1}}{\sqrt{\pi_{1:t}^2(i) + \gamma}} \right) \\
&\leq \sum_{t=1}^{T} \sum_{i=1}^{N} \frac{\pi_t^2(i)}{K} \cdot \left( \frac{c_t}{\sqrt{f_{1:t-1}^2(i) + \gamma}} - \frac{c_t}{\sqrt{\pi_{1:t}^2(i) + \gamma}} \right) && \triangleright c_t \leq c_{t+1} \\
&= \sum_{t=1}^{T} \sum_{i=1}^{N} \frac{\pi_t^2(i) \cdot c_t}{K \sqrt{\pi_{1:t}^2(i) + \gamma}} \cdot \left( \sqrt{1 + \frac{\pi_t^2(i)}{f_{1:t-1}^2(i) + \gamma}} - 1 \right) \\
&\leq \frac{c_T}{2K} \sum_{t=1}^{T} \sum_{i=1}^{N} \frac{\pi_t(i)^4}{\sqrt{\pi_{1:t}^2(i) + \gamma} \cdot (f_{1:t-1}^2(i) + \gamma)} && \triangleright \sqrt{1+x} - 1 \leq \frac{x}{2}
\end{aligned}
$$

Moreover, we observe that $\pi_{1:t}^2(i) \leq f_{1:t-1}^2(i) + \gamma$ and $\sqrt{\pi_{1:t}^2(i)} \leq \sqrt{\pi_{1:t}^2(i) + \gamma}$. Letting $\gamma = G^2$, and following Lemma 13 in (Borsos et al., 2018), we conclude this bound,

$$
\begin{aligned}
\sum_{t=1}^{T} (\ell_t(p^t) - \ell_t(p^{t+1})) &\leq \frac{c_T}{2K} \sum_{t=1}^{T} \sum_{i=1}^{N} \frac{\pi_t(i)^4}{(f_{1:t}^2(1))^{\frac{3}{2}}} \\
&= \sqrt{L} \cdot \frac{c_T}{2K} \sum_{t=1}^{T} \sum_{i=1}^{N} \frac{\pi_t(i)^4 / L^4}{(f_{1:t}^2(1)/G^2)^{\frac{3}{2}}} \\
&\leq \left( 22N\sqrt{L} \cdot \sum_{i=1}^{N} \sqrt{f_{1:T-1}^2(i) + G^2} \right) / K \\
&\leq \left( 22N\sqrt{L} \cdot \sum_{i=1}^{N} \sqrt{\pi_{1:T}^2(i)} + 22N^2 G^2 \right) / K
\end{aligned}
\tag{37}
$$

Then, we can get the final bound of (A) by plugging Equation equation 37 into Equation equation 36 and summarizing as follows,

$$
\sum_{t=1}^{T} \ell_t(p^t) - \sum_{t=1}^{T} \ell_t(p) \leq \frac{22NG}{K} \cdot \sum_{i=1}^{N} \sqrt{\pi_{1:T}^2(i)} + \frac{22N^2 G^2}{K} + \frac{NG^2}{p_{\min}}.
$$

**Bounding (B)**. Letting $a_i = \sqrt{\pi_{1:T}^2(i)}$ and combining Corollaries B.7, B.8 and B.9, we bound the term (B) as follows,

$$
\begin{aligned}
&\min_{p \in \Delta} \sum_{t=1}^{T} \ell_t(p) - \min_{p} \sum_{t=1}^{T} \ell_t(p) \\
&\leq \frac{K(\sum_{i=1}^{N} a_i)^2}{(K - Np_{\min})^2} - \frac{(\sum_{i=1}^{N} a_i)^2}{K} \\
&\leq \left( \frac{K}{(K - Np_{\min})^2} - \frac{1}{K} \right) \cdot \left( \sum_{i=1}^{N} \sqrt{\pi_{1:T}^2(i)} \right)^2 \\
&\leq \frac{6Np_{\min}}{K^2} \cdot \left( \sum_{i=1}^{N} \sqrt{\pi_{1:T}^2(i)} \right)^2
\end{aligned}
\tag{38}
$$

In the last line, we use the fact that $\frac{1}{(1-x)^2} - 1 \le 6x$ for $x \in [0, 1/2]$. Hence, we scale the coefficient

$$\frac{K}{(K - Np_{\min})^2} - \frac{1}{K} = \frac{1}{K}\Big[\frac{1}{(1 - Np_{\min}/K)^2} - 1\Big] \le \frac{6Np_{\min}}{K^2},$$

where we let $p_{\min} \le K/(2N)$.

**Summary**. Setting $\gamma = G^2$, and combining the bound in Equation equation 36 and Equation equation 38, we have,

$$\begin{aligned}
N^2 \cdot \text{Regret}_{\text{FTRL}}(T) &= \sum_{t=1}^{T} \ell_t(p^t) - \min_p \sum_{t=1}^{T} \ell_t(p) \\
&\le \frac{22NG}{K} \cdot \sum_{i=1}^{N} \sqrt{\pi_{1:T}^2(i)} + \frac{22N^2G^2}{K} + \frac{NG^2}{p_{\min}} + \frac{6Np_{\min}}{K^2} \cdot \Big(\sum_{i=1}^{N} \sqrt{\pi_{1:T}^2(i)}\Big)^2.
\end{aligned} \tag{39}$$

The $p_{\min}$ is only relevant for the theoretical analysis. Hence, the choice of it is arbitrary, and we can set it to $p_{\min} = \min\Big\{K/(2N), GK/(\sqrt{6}\sum_{i=1}^{N} \sqrt{\pi_{1:T}^2(i)})\Big\}$ which turns the upper bound to the minimal value. Hence, we yield the final bound of FTRL in the end,

$$\text{Regret}_{\text{S}}(T) \le \frac{27G}{NK} \sum_{i=1}^{N} \sqrt{\pi_{1:T}^2(i)} + \frac{22G^2}{K} \tag{40}$$

$\square$

## C.3 EXPECTED REGRET OF PARTIAL FEEDBACK: PROOF OF THEOREM 3.4

*Proof.* Using the property of unbiasedness, we have

$$\begin{aligned}
&\frac{1}{N^2} \min_p \mathbb{E}[\sum_{t=1}^{T} \ell_t(\tilde{p}^t) - \sum_{t=1}^{T} \ell_t(p)] \\
&= \frac{1}{N^2} \min_p \mathbb{E}[\sum_{t=1}^{T} \tilde{\ell}_t(\tilde{p}^t) - \sum_{t=1}^{T} \tilde{\ell}_t(p)] \\
&= \underbrace{\frac{1}{N^2} \mathbb{E}\Big[\sum_{t=1}^{T} \tilde{\ell}_t(\tilde{p}^t) - \sum_{t=1}^{T} \tilde{\ell}_t(p^t)\Big]}_{(A)} + \underbrace{\frac{1}{N^2} \min_p \mathbb{E}\Big[\sum_{t=1}^{T} \tilde{\ell}_t(p^t) - \sum_{t=1}^{T} \ell_t(p)\Big]}_{(B)}.
\end{aligned} \tag{41}$$

**Bounding (A)**. We recall that $\tilde{p}_i^t \ge \frac{\theta K}{N}$ for all $t \in [T], i \in [N]$ due to the mixing. Meanwhile, $p_i^t \ge K/N$ implies $\tilde{p}_i^t \ge K/N$. Thus, we have

$$\frac{1}{\tilde{p}_i^t} - \frac{1}{p_i^t} = \theta \cdot \frac{p_i^t - \frac{K}{N}}{\tilde{p}_i^t p_i^t} \le \theta \cdot \frac{p_i^t}{\tilde{p}_i^t p_i^t} = \frac{\theta}{\tilde{p}_i^t} \le \theta \cdot \frac{N}{K}.$$

Moreover, if $p_i^t \le K/N$, the above inequality still holds. We extend the (A) as follows,

$$\begin{aligned}
N^2 \cdot (A) &:= \mathbb{E}\Big[\sum_{t=1}^{T} \tilde{\ell}_t(\tilde{p}^t) - \sum_{t=1}^{T} \tilde{\ell}_t(p^t)\Big] \\
&= \mathbb{E}\Big[\sum_{t=1}^{T} \sum_{i=1}^{N} \tilde{\pi}_t^2(i)\Big(\frac{1}{\tilde{p}_i^t} - \frac{1}{p_i^t}\Big)\Big] \\
&\le \theta \cdot \frac{N}{K} \cdot \mathbb{E}\Big[\sum_{t=1}^{T} \sum_{i=1}^{N} \tilde{\pi}_t^2(i)\Big] \\
&\le \frac{\theta G^2 T N^2}{K},
\end{aligned}$$

where we use $\mathbb{E}[\tilde{\pi}_t^2(i)] = \pi_t^2(i) \le G^2$.

**Bounding (B)**. We note that $p^t$ is the decision sequence playing FTRL with the mixed cost functions. Thus, we combine the mixing bound of feedback (i.e., $\tilde{\pi}_t^2(i) \le \frac{G^2 N}{\theta K}$) and Theorem C.1. Replacing $G^2$ with $\frac{G^2 N}{\theta K}$, we get

$$\frac{1}{N^2}\left(\sum_{t=1}^{T}\tilde{\ell}_t(p^t) - \min_p \sum_{t=1}^{T}\tilde{\ell}_t(p)\right) \le \frac{27G}{\sqrt{\theta N K^3}} \cdot \mathbb{E}\left[\sum_{i=1}^{N}\sqrt{\tilde{\pi}_{1:T}^2(i)}\right] + \frac{22G^2 N}{\theta K^2}.$$

**Summary**. Using Jensen's inequality, we have $\mathbb{E}\left[\sum_{i=1}^{N}\sqrt{\tilde{\pi}_{1:T}^2(i)}\right] \le \sum_{i=1}^{N}\sqrt{\mathbb{E}[\tilde{\pi}_{1:T}^2(i)]} = \sum_{i=1}^{N}\sqrt{\pi_{1:T}^2(i)}$. Finally, we can get the upper bound of the regret in partial-bandit feedback,

$$\frac{1}{N^2}\min_p \mathbb{E}[\sum_{t=1}^{T}\ell_t(\tilde{p}^t) - \sum_{t=1}^{T}\ell_t(p)] \le \frac{\theta G^2 T}{K} + \frac{27G}{\sqrt{\theta K^3 N}} \cdot \sum_{i=1}^{N}\sqrt{\pi_{1:T}^2(i)} + \frac{22NG^2}{\theta K^2}. \quad (42)$$

Note that we can optimize the upper bound of regret in terms of $\theta$. Besides, $\theta$ is independent on $T$. Using the bound $\sum_{i=1}^{N}\sqrt{\pi_{1:T}^2(i)} \le NG\sqrt{T}$, we set $\theta = (\frac{N}{TK})^{\frac{1}{3}}$ to get the minimized bound. Additionally, we are pursuing an expected regret, which is $\text{Regret}_{(S)}(T)$ in the original definition in Equation equation 7. Using the unbiasedness of the mixed estimation and modified costs, we can obtain the final bound:

$$\begin{aligned}
\mathbb{E}[\text{Regret}_{(S)}(T)] &= \mathbb{E}[\sum_{t=1}^{T}\ell_t(\tilde{p}^t) - \min_p \sum_{t=1}^{T}\ell_t(p)] \\
&= \mathbb{E}[\sum_{t=1}^{T}\ell_t(\tilde{p}^t) - \min_p \sum_{t=1}^{T}\tilde{\ell}_t(p)] + \mathbb{E}[\min_p \sum_{t=1}^{T}\tilde{\ell}_t(p) - \min_p \sum_{t=1}^{T}\ell_t(p)] \\
&\le \mathcal{O}\big(N^{\frac{1}{3}}T^{\frac{2}{3}}/K^{\frac{4}{3}}\big) + \mathbb{E}[\min_p \sum_{t=1}^{T}\tilde{\ell}_t(p) - \min_p \sum_{t=1}^{T}\ell_t(p)] \\
&\le \tilde{\mathcal{O}}\big(N^{\frac{1}{3}}T^{\frac{2}{3}}/K^{\frac{4}{3}}\big),
\end{aligned}$$

where the last inequality uses the conclusion in Theorem 8 (Borsos et al., 2018), which induces an additional log term. □

## D   DETAILS PROOFS OF CONVERGENCE GUARANTEES

We start our convergence analysis with a clarification of the concepts of optimal independent sampling. Considering an Oracle always outputs the optimal probabilities $p^*$, we have,

$$\delta_*^t = \mathbb{E}\left[\left\|\sum_{i\in S^*}\frac{\lambda_i g_i^t}{p_i^*} - \sum_{i=1}^{N}\lambda_i g_i^t\right\|^2\right] = \mathbb{E}\left[\sum_{i=1}^{N}\frac{1-p_i^*}{p_i^*}\|\tilde{g}_i^t\|^2\right],$$

where we have $\|\tilde{g}_i^t\|^2 = \|\lambda_i g_i^t\|^2$.

Then, we plug the optimal probability in Equation 27 into the above equation to obtain

$$\delta_*^t = \mathbb{E}\left[\sum_{i=1}^{N}\frac{1-p_i^*}{p_i^*}\|\tilde{g}_i^t\|^2\right] = \mathbb{E}\left[\frac{1}{K-(N-l)}\left(\sum_{i=1}^{l}\|\tilde{g}_i^t\|\right)^2 - \sum_{i=1}^{l}\|\tilde{g}_i^t\|^2\right].$$

Using the fact that $K\|\tilde{g}_N^t\| \le \sum_{i=1}^N \|\tilde{g}_i^t\|$, we have

$$\delta_*^t \le \mathbb{E}\left[\frac{1}{K}\left(\sum_{i=1}^N \|\tilde{g}_i^t\|\right)^2 - \sum_{i=1}^N \|\tilde{g}_i^t\|^2\right]$$

$$= \mathbb{E}\left[\frac{1}{K}\left(\sum_{i=1}^N \|\tilde{g}_i^t\|\right)^2\left(1 - K\frac{\sum_{i=1}^N \|\tilde{g}_i^t\|^2}{\left(\sum_{i=1}^N \|\tilde{g}_i^t\|\right)^2}\right)\right]$$

$$\le \frac{N-K}{NK}\mathbb{E}\left[\left(\sum_{i=1}^N \|\tilde{g}_i^t\|\right)^2\right].$$

For an independent uniform sampling $S^t \sim \mathbb{U}(p_i = \frac{K}{N})$, we have

$$\delta^t := \mathbb{E}\left[\left\|\sum_{i \in S^t}\frac{\lambda_i}{p_i}g_i^t - \sum_{i=1}^N \lambda_i g_i^t\right\|^2\right] = \mathbb{E}\left[\sum_{i=1}^N \frac{1 - \frac{K}{N}}{\frac{K}{N}}\|\tilde{g}_i^t\|^2\right] = \frac{N-K}{K}\mathbb{E}\left[\sum_{i=1}^N \|\tilde{g}_i^t\|^2\right]$$

For a uniform random sampling $S^t \sim \mathbb{U}(p_i = \frac{K}{N})$, we have

$$\delta_{\mathbb{U}} := \mathbb{E}\left[\left\|\sum_{i \in S^t}\frac{\lambda_i}{p_i}g_i^t - \sum_{i=1}^N \lambda_i g_i^t\right\|^2\right] \le \frac{N-K}{N-1}\frac{N}{K}\mathbb{E}\left[\sum_{i=1}^N \|\tilde{g}_i^t\|^2\right]. \tag{43}$$

Putting Equations together induces the improvement factor of optimal independent sampling respecting uniform random sampling:

$$\alpha_*^t := \frac{\delta_*^t}{\delta_{\mathbb{U}}} = \frac{\mathbb{E}\left[\left\|\sum_{i \in S^*}\frac{\lambda_i}{p_i^*}g_i^t - \sum_{i=1}^N \lambda_i g_i^t\right\|^2\right]}{\mathbb{E}\left[\left\|\sum_{i \in S^t}\frac{\lambda_i}{p_i}g_i^t - \sum_{i=1}^N \lambda_i g_i^t\right\|^2\right]} \le \frac{(N-1)\mathbb{E}\left[\left(\sum_{i=1}^N \|\tilde{g}_i^t\|\right)^2\right]}{N^2\mathbb{E}\left[\sum_{i=1}^N \|\tilde{g}_i^t\|^2\right]} < \frac{\mathbb{E}\left[\left(\sum_{i=1}^N \|\tilde{g}_i^t\|\right)^2\right]}{N\mathbb{E}\left[\sum_{i=1}^N \|\tilde{g}_i^t\|^2\right]} \le 1. \tag{44}$$

Now we are ready to give our convergence analysis in detail.

*Proof.* We recall the updating rule during round $t$ as:

$$x^{t+1} = x^t - \eta_g \sum_{i \in S^t}\frac{\lambda_i g_i^t}{p_i^t} = x^t - \eta_g d^t, \text{ where } g_i^t = x^t - x_i^{t,R} = \eta_l \sum_{r=1}^R \nabla F_i(x_i^{t,r-1}).$$

**Notation.** For clear notation, we denote $W = \max\{\lambda_i\}_{i \in [N]}, \gamma_*^t = \frac{(N-K)\alpha_*^t + K}{K}$.

**Upper bound of local drift**. We need the upper bound of local drift at first. For $r \in [R]$, we have

$$\mathbb{E}\left[\|g_i^t\|^2\right] = \mathbb{E}\|x_i^{t,r} - x^t\|^2 = \mathbb{E}\left\|x_i^{t,r-1} - x^t - \eta_l \nabla F_i(x_i^{t,r-1})\right\|^2$$

$$= \mathbb{E}\left\|x_i^{t,r-1} - x^t - \eta_l\left(\nabla F_i(x_i^{t,r-1}) \pm \nabla f_i\left(x_i^{t,r-1}\right) \pm \nabla f_i\left(x^t\right)\right)\right\|^2$$

$$\le \left(1 + \frac{1}{2R-1}\right)\mathbb{E}\left\|x_i^{t,r-1} - x^t\right\|^2 + 3\mathbb{E}\left\|\eta_l\left(\nabla F_i(x_i^{t,r-1}) - \nabla f_i\left(x_i^{t,r-1}\right)\right)\right\|^2$$

$$+ 3\mathbb{E}\left[\left\|\eta_l\left(\nabla f_i\left(x_i^{t,r-1}\right) - \nabla f_i\left(x^t\right)\right)\right\|^2\right] + 3\mathbb{E}\left[\left\|\eta_l\left(\nabla f_i\left(x^t\right)\right)\right\|^2\right]$$

$$\le \left(1 + \frac{1}{2R-1} + 3\eta_l^2 L^2\right)\mathbb{E}\left\|x_i^{t,r-1} - x^t\right\|^2 + \eta_l^2(\sigma_l^2 + 3\left\|\nabla f_i\left(x^t\right)\right\|^2)$$

Unrolling the recursion, we obtain

$$
\begin{aligned}
\mathbb{E}\left\|x_i^{t,r} - x^t\right\|^2 &\leq \sum_{p=0}^{r-1}\left(1 + \frac{1}{2R-1} + 3\eta_l^2 L^2\right)^p \eta_l^2 \left(\sigma_l^2 + 3\left\|\nabla f_i\left(x^t\right)\right\|^2\right) \\
&\leq (R-1)\left[\left(1 + \frac{1}{R-1}\right)^R - 1\right]\eta_l^2\left(\sigma_l^2 + 3\left\|\nabla f_i\left(x^t\right)\right\|^2\right) \\
&\leq 5R\eta_l^2\left(\sigma_l^2 + 3\left\|\nabla f_i\left(x^t\right)\right\|^2\right) \leq \eta_l^2\left(\sigma_l^2 + 3\left\|\nabla f_i\left(x^t\right)\right\|^2\right),
\end{aligned}
\tag{45}
$$

where we use the fact that $(1 + \frac{1}{R-1})^R \leq 5$ for $R > 1$ and replace $\eta_l \leq \frac{1}{\sqrt{5R}}\eta_l$. Therefore, we have

$$
\sum_{i=1}^{N}\lambda_i^2\|g_i^t\|^2 \leq \sum_{i=1}^{N}\lambda_i^2\eta_l^2\left(\sigma_l^2 + 3\left\|\nabla f_i\left(x^t\right)\right\|^2\right) \leq W\eta_l^2(\sigma_l^2 + 3\sigma_g^2 + 3\|f(x^t)\|^2),
\tag{46}
$$

where we use the fact by Assumption 3.7 that $\sum_{i=1}^{N}\lambda_i\|\nabla f_i(x^t)\|^2 \leq \|\nabla f(x^t)\|^2 + \sigma_g^2$.

**Descent lemma.** Using the smoothness of $f$ and taking expectations conditioned on $x$ and over the sampling $S^t$, we have

$$
\begin{aligned}
\mathbb{E}\left[f(x^{t+1})\right] = f(x^t - \eta_g d^t) &\leq f(x^t) - \eta_g\left\langle\nabla f(x^t), d^t\right\rangle + \frac{L}{2}\eta_g^2\mathbb{E}\left[\|d^t\|^2\right] \\
&\leq f(x^t) - \eta_g\|\nabla f(x^t)\|^2 + \eta_g\left\langle\nabla f(x^t), \nabla f(x^t) - d^t\right\rangle + \frac{L}{2}\eta_g^2\mathbb{E}\left[\|d^t\|^2\right] \\
&\leq f(x^t) - \frac{\eta_g}{2}\|\nabla f(x^t)\|^2 + \frac{\eta_g}{2}\mathbb{E}\left[\|\nabla f(x^t) - d^t\|^2\right] + \frac{L}{2}\eta_g^2\mathbb{E}\left[\|d^t\|^2\right],
\end{aligned}
\tag{47}
$$

where the last inequality follows since $\langle a, b\rangle \leq \frac{1}{2}\|a\|^2 + \frac{1}{2}\|b\|^2, \forall a, b \in \mathbb{R}^d$.

We first investigate the expectation gap between global first-order gradient and utilized global estimates,

$$
\mathbb{E}\left[\left\|\nabla f(x^t) - d^t\right\|^2\right] = \mathbb{E}\left[\left\|\sum_{i=1}^{N}\lambda_i \nabla f_i(x^t) - \sum_{i=1}^{N}\lambda_i g_i^t\right\|^2\right] = \mathbb{E}\left[\left\|\sum_{i=1}^{N}\lambda_i\left(\nabla f_i(x^t) - g_i^t\right)\right\|^2\right]
$$

$$
\leq 2N\sum_{i=1}^{N}\lambda_i^2 \mathbb{E}\left[\left\|\frac{1}{R}\sum_{r=1}^{R}\nabla f_i(x^t) - \eta_l\sum_{r=1}^{R}\nabla f_i(x_i^{t,r-1})\right\|^2\right]
$$

$$
+ 2N\sum_{i=1}^{N}\lambda_i^2 \mathbb{E}\left[\left\|\eta_l\sum_{r=1}^{R}\nabla f_i(x^{t,r-1}) - \eta_l\sum_{r=1}^{R}\nabla F_i(x_i^{t,r-1})\right\|^2\right]
$$

$$
\leq \frac{2N}{R^2}\sum_{i=1}^{N}\lambda_i^2 \mathbb{E}\left[\left\|\sum_{r=1}^{R}\nabla f_i(x^t) - \eta_l R\sum_{r=1}^{R}\nabla f_i(x_i^{t,r-1})\right\|^2\right] + 2\eta_l^2 R^2\sigma_l^2
$$

$$
= \frac{2N}{R^2}\sum_{i=1}^{N}\lambda_i^2 \mathbb{E}\left[\left\|(1-\eta_l R)\sum_{r=1}^{R}\nabla f_i(x^t) + \eta_l R\left(\sum_{r=1}^{R}\nabla f_i(x^t) - \sum_{r=1}^{R}\nabla f_i(x_i^{t,r-1})\right)\right\|^2\right] + 2\eta_l^2 R^2\sigma_l^2
$$

$$
\leq \frac{4N}{R^2}\sum_{i=1}^{N}\lambda_i^2 \mathbb{E}\left[\left\|(1-\eta_l R)\sum_{r=1}^{R}\nabla f_i(x^t)\right\|^2\right]
$$

$$
+ \frac{4N}{R^2}\sum_{i=1}^{N}\lambda_i^2 \mathbb{E}\left[\left\|\eta_l R\left(\sum_{r=1}^{R}\nabla f_i(x^t) - \sum_{r=1}^{R}\nabla f_i(x_i^{t,r-1})\right)\right\|^2\right] + 2\eta_l^2 R^2\sigma_l^2
$$

$$
\leq 4N(1-\eta_l R)^2\sum_{i=1}^{N}\lambda_i^2 \mathbb{E}\left[\left\|\nabla f_i(x^t)\right\|^2\right]
$$

$$
+ 4N\eta_l^2\sum_{i=1}^{N}\lambda_i^2 R\sum_{r=1}^{R}\mathbb{E}\left[\left\|\nabla f_i(x^t) - \nabla f_i(x_i^{t,r-1})\right\|^2\right] + 2\eta_l^2 R^2\sigma_l^2
$$

$$
\leq 4N(1-\eta_l R)^2\sum_{i=1}^{N}\lambda_i^2 \mathbb{E}\left[\left\|\nabla f_i(x^t)\right\|^2\right] + 4N\eta_l^2 L^2\sum_{i=1}^{N}\lambda_i^2 R\sum_{r=1}^{R}\mathbb{E}\left[\left\|x^t - x^{t,r-1}\right\|^2\right] + 2\eta_l^2 R^2\sigma_l^2.
$$

$$
\leq 4N(1-\eta_l R)^2\sum_{i=1}^{N}\lambda_i^2 \mathbb{E}\left[\left\|\nabla f_i(x^t)\right\|^2\right] + 4N\eta_l^2 L^2 R^2\sum_{i=1}^{N}\lambda_i^2 \mathbb{E}\left[\left\|g_i^t\right\|^2\right] + 2\eta_l^2 R^2\sigma_l^2.
$$

$$
\leq 4N(1-\eta_l R)^2 W(\mathbb{E}\left[\left\|\nabla f(x^t)\right\|^2\right] + \sigma_g^2) + 4N\eta_l^4 L^2 R^2 W\left(\sigma_l^2 + 3\sigma_g^2 + 3\mathbb{E}\left[\left\|\nabla f(x^t)\right\|^2\right]\right) + 2\eta_l^2 R^2\sigma_l^2,
$$

$$(48)$$

where we plug equation 46 at the last. If we replace $\eta_l \leq \frac{1}{R}\eta_l$ and use the fact that $W$ is proportional to $\frac{1}{N}$ (omit factor $NW$), we have

$$
\frac{\eta_g}{2}\mathbb{E}\left[\left\|\nabla f(x^t) - d^t\right\|^2\right]
$$

$$
\leq 2\eta_g(1-\eta_l R)^2(\mathbb{E}\left[\left\|\nabla f(x^t)\right\|^2\right] + \sigma_g^2) + 2\eta_g\eta_l^2 L^2\left(\sigma_l^2 + 3\sigma_g^2 + 3\mathbb{E}\left[\left\|\nabla f(x^t)\right\|^2\right]\right) + \eta_g\eta_l^2\sigma_l^2
$$

$$
\leq \left(2\eta_g(1-\eta_l R)^2 + 6\eta_g\eta_l^2 L^2\right)\mathbb{E}\left[\left\|\nabla f(x^t)\right\|^2\right]
$$

$$
+ \left(2\eta_g(1-\eta_l R)^2 + 6\eta_g\eta_l^2 L^2\right)\sigma_g^2 + \eta_g\eta_l^2\left(2L^2 + 1\right)\sigma_l^2
$$

$$(49)$$

Now, we focus on the quality of estimates,

$$
\mathbb{E}\left[\|d^t\|^2\right] \leq \mathbb{E}\left[\left\|\sum_{i \in S^t} \frac{\lambda_i g_i^t}{p_i^t} - \sum_{i=1}^N \lambda_i g_i^t\right\|^2 + \left\|\sum_{i=1}^N \lambda_i g_i^t\right\|^2\right]
$$

$$
\leq \underbrace{\mathbb{E}\left[\left\|\sum_{i \in S^t} \frac{\lambda_i g_i^t}{p_i^t} - \sum_{i \in S^*} \frac{\lambda_i g_i^t}{p_i^*}\right\|^2\right]}_{Q(S^t)} + \underbrace{\mathbb{E}\left[\left\|\sum_{i \in S^*} \frac{\lambda_i g_i^t}{p_i^*} - \sum_{i=1}^N \lambda_i g_i^t\right\|^2\right] + \mathbb{E}\left[\left\|\sum_{i=1}^N \lambda_i g_i^t\right\|^2\right]}_{(A)}.
$$

(50)

Here, we note that $Q(S^t)$ is the main point in this paper. The term $(A)$ indicates the intrinsic gap for a sampling Oracle to approach its targets and the quality of the targets for optimization. Using definition in equation 43 and equation 44, we have

$$
(A) = \mathbb{E}\left[\left\|\sum_{i \in S^*} \frac{\lambda_i g_i^t}{p_i^*} - \sum_{i=1}^N \lambda_i g_i^t\right\|^2\right] + \mathbb{E}\left[\left\|\sum_{i=1}^N \lambda_i g_i^t\right\|^2\right]
$$

$$
\leq \alpha_*^t \frac{(N-K)N}{(N-1)K} \mathbb{E}\left[\sum_{i=1}^N \lambda_i^2 \|g_i^t\|^2\right] + \mathbb{E}\left[\left\|\sum_{i=1}^N \lambda_i g_i^t\right\|^2\right]
$$

$$
\leq \alpha_*^t \frac{(N-K)N}{(N-1)K} \sum_{i=1}^N \lambda_i^2 \mathbb{E}\left[\|g_i^t\|^2\right] + N \sum_{i=1}^N \lambda_i^2 \mathbb{E}\left[\|g_i^t\|^2\right]
$$

$$
= \left(\alpha_*^t \frac{(N-K)N}{(N-1)K} + N\right) \sum_{i=1}^N \lambda_i^2 \mathbb{E}\left[\|g_i^t\|^2\right]
$$

$$
\leq \left(\frac{\alpha_*^t(N-K)+K}{K}\right) \frac{N}{N-1} \sum_{i=1}^N \lambda_i^2 \mathbb{E}\left[\|g_i^t\|^2\right]
$$

$$
= \gamma_*^t \frac{N}{N-1} \sum_{i=1}^N \lambda_i^2 \mathbb{E}\left[\|g_i^t\|^2\right]
$$

$$
\leq \gamma_*^t \frac{N}{N-1} W(\sigma_l^2 + 3\sigma_g^2 + 3\|f(x^t)\|^2)
$$

where $\gamma_*^t := \frac{\alpha_*^t(N-K)+K}{K} \in [1, \frac{N}{K}]$ as we defined before. Then, we have

$$
\frac{L}{2}\eta_g^2 \mathbb{E}\left[\|d^t\|^2\right] \leq \frac{L}{2}\eta_g^2 Q(S^t) + \frac{L}{2}\eta_g^2 (A)
$$

$$
\leq \frac{L}{2}\eta_g^2 Q(S^t) + \frac{L}{2}\eta_g^2 \eta_l^2 \gamma_*^t \frac{N}{N-1} W(\sigma_l^2 + 3\sigma_g^2 + 3\|f(x^t)\|^2)
$$

(51)

$$
\leq \frac{L}{2}\eta_g^2 Q(S^t) + L\eta_g^2 \eta_l^2 \gamma_*^t W(\sigma_l^2 + 3\sigma_g^2 + 3\|f(x^t)\|^2),
$$

where we let $\frac{N}{2(N-1)} \leq 1$ the last inequality for simplicity of notation.

**Putting together.** Substituting corresponding terms in equation 47 with equation 49 and equation 51 to finish the descent lemma, we have

$$
\mathbb{E}\left[f(x^{t+1})\right] \leq f(x^t) + \frac{L}{2}\eta_g^2 Q(S^t) - \frac{\eta_g}{2}\|\nabla f(x^t)\|^2
$$

$$
+ 2\eta_g(1-\eta_l R)^2 (\mathbb{E}\left[\|\nabla f(x^t)\|^2\right] + \sigma_g^2) + 2\eta_g\eta_l^2 L^2 \left(\sigma_l^2 + 3\sigma_g^2 + 3\mathbb{E}\left[\|\nabla f(x^t)\|^2\right]\right) + \eta_g\eta_l^2 \sigma_l^2
$$

$$
+ L\eta_g^2\eta_l^2 \gamma_*^t W(\sigma_l^2 + 3\sigma_g^2 + 3\|f(x^t)\|^2).
$$

(52)

Then, we rearrange the terms to obtain

$$
\begin{aligned}
\mathbb{E}\left[f(x^{t+1})\right] \leq f(x^t) &+ \frac{L}{2}\eta_g^2 Q(S^t) - \frac{\eta_g}{2}\left(1 - 4(1-\eta_l R)^2 - 12\eta_l^2 L^2 - 6L\eta_g\eta_l^2\gamma_*^t W\right)\|\nabla f(x^t)\|^2 \\
&+ \left(2\eta_g(1-\eta_l R)^2 + 6\eta_g\eta_l^2 L^2\right)\sigma_g^2 + \eta_g\eta_l^2\left(2L^2+1\right)\sigma_l^2 \\
&+ L\eta_g^2\gamma_*^t W(\sigma_l^2 + 3\sigma_g^2).
\end{aligned}
\tag{53}
$$

Taking a full expectation on both side and rearranging equation 52 and setting $\eta_g \leq \frac{1}{L}$ to adapt $L$, we obtain

$$
\rho^t\mathbb{E}\|\nabla f(x^t)\|^2 \leq \frac{2(\mathbb{E}[f(x^t)] - \mathbb{E}[f(x^{t+1})])}{\eta_g} + \beta^t\eta_g + \epsilon + Q(S^t),
\tag{54}
$$

where we define

$$
\begin{aligned}
\rho^t &:= \left(1 - 4(1-\eta_l R)^2 - 12\eta_l^2 L^2 - 6\eta_l^2\gamma_*^t W\right), \\
\beta^t &:= 2L\gamma_*^t W(\sigma_l^2 + 3\sigma_g^2), \\
\epsilon &= 4\left((1-\eta_l R)^2 + 3\eta_l^2 L^2\right)\sigma_g^2 + 2\eta_l^2\left(2L^2+1\right)\sigma_l^2.
\end{aligned}
$$

Taking averaging of both sides of Equation 54 over from time $1$ to $T$, we have

$$
\frac{1}{T}\sum_{t=1}^{T}\rho^t\mathbb{E}\|\nabla f(x^t)\|^2 \leq \frac{2(\mathbb{E}\left[f(x^1) - f(x^T)\right])}{T\eta_g} + \bar{\beta}\eta_g + \sum_{t=1}^{T}\frac{Q(S^t)}{T} + \epsilon,
$$

where $\bar{\beta} = \frac{1}{T}\sum_{t=1}^{T}\beta^t$. Then, taking upper bound $\mathbb{E}\left[f(x^1) - f(x^{+\infty})\right] \leq M$, $\hat{\rho} := \min\{\rho^t\}_{t=1}^{T}$, setting $\eta_g = \sqrt{\frac{2M}{T\bar{\beta}}}$ to minimize the upper bound, $\eta_l \leq \min(\frac{1}{R}, \frac{1}{\sqrt{5}R})$, we have

$$
\min_{t\in[T]}\mathbb{E}\|\nabla f(x^t)\|^2 \leq \frac{1}{T}\sum_{t=1}^{T}\frac{\rho^t}{\hat{\rho}}\mathbb{E}\|\nabla f(x^t)\|^2 \leq \sqrt{\frac{8M\bar{\beta}}{T\hat{\rho}^2}} + \frac{\frac{1}{T}\sum_{t=1}^{T}Q(S^t) + \epsilon}{\hat{\rho}},
$$

which concludes the proof. $\qquad\square$

# E  FURTHER DISCUSSIONS

## E.1  A SKETCH OF PROOF WITH CLIENT STRAGGLERS

We note the possibility that some clients are unavailable to participants due to local failure or being busy in each round. To extend our analysis to the case, we assume there is a known distribution of client availability $\mathcal{A}$ such that a subset $\mathcal{A}^t \sim \mathcal{A}$ of clients are available at the $t$-th communication round. Let $q_i = \text{Prob}(i \in \mathcal{A}^t)$ denote the probability that client $i$ is available at round $t$. Based on the setting, we update the definition of estimation $g^t$:

$$
g^t := \sum_{i\in S^t\subseteq\mathcal{A}^t}\frac{\lambda_i g_i^t}{q_i p_i^t},
$$

where $S^t \subseteq \mathcal{A}^t$ indicates that we can only sample from available set. Then, we apply the estimation to variance and obtain the following target:

$$
\text{Regret}(T) = \frac{1}{N^2}\left(\sum_{t=1}^{T}\sum_{i=1}^{N}\frac{\pi_t^2(i)}{q_i p_i} - \sum_{t=1}^{T}\min_p\sum_{i=1}^{N}\frac{\pi_t^2(i)}{q_i p_i}\right).
$$

Analogous to our analysis in Appendix C, we could obtain a similar bound of the above regret that takes the availability into consideration.

## E.2  DIFFERENCES BETWEEN BIASED CLIENT SAMPLING METHODS

This section discusses the main differences between unbiased client sampling and biased client sampling methods. The proposed K-Vib sampler is an unbiased sampler for the first-order gradient

of objective 1. Recent biased client sampling methods include Power-of-Choice (POC) (Cho et al., 2020b) and DivFL (Balakrishnan et al., 2022). Concretely, POC requires all clients to upload local empirical loss as prior knowledge and selects clients with the largest empirical loss. DivFL builds a submodular based on the latest gradient from clients and selects clients to approximate all client information. Therefore, these client sampling strategies build a biased gradient estimation that may deviate from a fixed global goal.

FL with biased client sampling methods, such as POC and DivFL, can be considered dynamic re-weighting algorithms adjusting $p_i$. Analogous to the Equation 1, the basic objective of FL with biased client sampling methods can be defined as follows (Li et al., 2020; Balakrishnan et al., 2022; Cho et al., 2020b):

$$\min_{x \in \mathcal{X}} f(x) := \sum_{i=1}^{N} p_i f_i(x) := \sum_{i=1}^{N} p_i \mathbb{E}_{\xi_i \sim \mathcal{D}_i}[F_i(x, \xi_i)], \tag{55}$$

where $p$ is the probability simplex, and $p_i$ is the probability of client $i$ being sampled. The gradient estimation is defined as $g^t = \frac{1}{K} \sum_{i \in S^t} g_i$ accordingly. The targets of biased FL client sampling are determined by the sampling probability $p$ as a replacement of $\lambda$ in the original FedAvg objective 1. Typically, the value of $p$ is usually dynamic and implicit.

### E.3 THEORETICAL COMPARISON WITH OSMD

The K-Vib sampler proposed in this paper is orthogonal with the recent work OSMD sampler Zhao et al. (2022)[4] in theoretical contribution. We justify our points below:

a) According to Equations (6) and (7) in OSMD, it proposes an online mirror descent procedure that optimizes the additional estimates to replace the mixing strategy in Vrb Borsos et al. (2018). The approach can be also utilized as an alternative method in Equation 9.

b) The improvement of the K-Vib sampler is obtained from the modification of the sampling procedure. In contrast, the OSMD still follows the conventional random sampling procedure, as we discussed in Lemma 2.2. Hence, our theoretical findings of applying the independent sampling procedure in adaptive client sampling can be transferred to OSMD as well.

In short, the theoretical improvement of our work is different from the OSMD sampler. And, our insights about utilizing the independent sampling procedure can be used to improve the OSMD sampler. Meanwhile, the OSMD also suggests future work for the K-Vib sampler in optimizing the additional estimates procedure instead of mixing.

## F EXPERIMENTS DETAILS

**Distribution of Datasets**. The data distribution across clients is shown in Figure 6. The task setting follows the FL literature (Li et al., 2020; Chen et al., 2020).

**Hyper-parameters Setting**. For all samplers, there is an implicit value $G$ (Lipschitz gradient) related to the hyper-parameters. We set $G = 0.01$ for the Synthetic dataset task and $G = 0.1$ for FEMNIST tasks. We set $\eta = 0.4$ for Mabs (Salehi et al., 2017) as suggested by the original paper. Vrb Borsos et al. (2018) also utilize mixing strategy $\theta = (N/T)^{\frac{1}{3}}$ and regularization $\gamma = G^2 * N/\theta$. For the case that $N > T$ in FEMNIST tasks, we set $\theta = 0.3$ following the official source code[5]. For Avare El Hanchi & Stephens (2020), we set $p_{\min} = \frac{1}{5N}$, $C = \frac{1}{\frac{1}{N} - p_{\min}}$ and $\delta = 1$ for constant-stepsize as suggested in Appendix D of original paper. For the K-Vib sampler, we set $\theta = (\frac{N}{TK})^{1/3}$ and $\gamma = G^2 \frac{N}{K\theta}$. We also fix $\gamma$ and $\theta = 0.3$ for our sensitivity study in Figure 3.

**Baselines with budget $K$.** Our theoretical results in Theorem 3.4 and empirical results in Figure 2 reveal a key improvement of our work, that is, the linear speed up in online convex optimization. In contrast, we provide additional experiments with the different budget $K$ in Figure 7. Baseline methods do not preserve the improvement property respecting large budget $K$ in adaptive client

---

[4]we refer to the latest version https://arxiv.org/pdf/2112.14332.pdf
[5]https://github.com/zalanborsos/online-variance-reduction

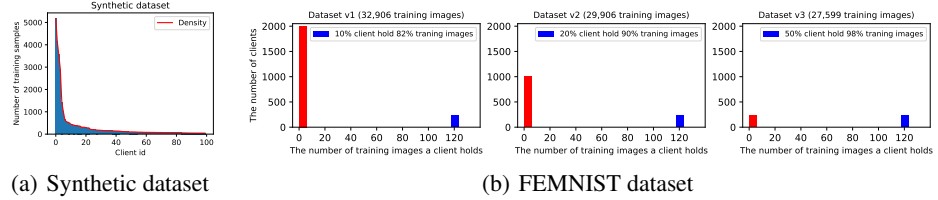

(a) Synthetic dataset          (b) FEMNIST dataset

Figure 6: Distributed of federated datasets.

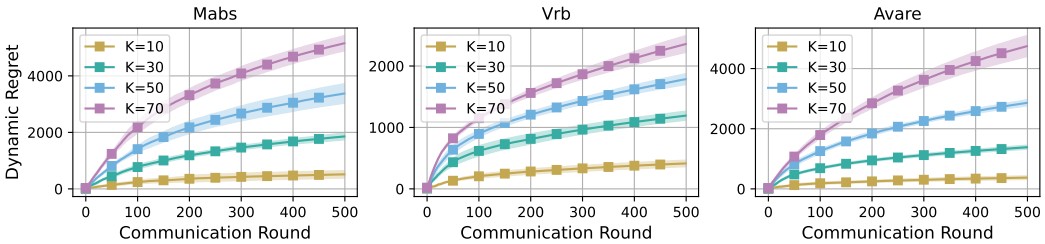

Figure 7: Regret of baseline algorithms with different $K$

sampling for variance reduction. Moreover, with the increasing communication budget $K$, the optimal sampling value is decreasing. As a result, the regret of baselines increases in Figure 7, indicating the discrepancy to the optimal is enlarged.

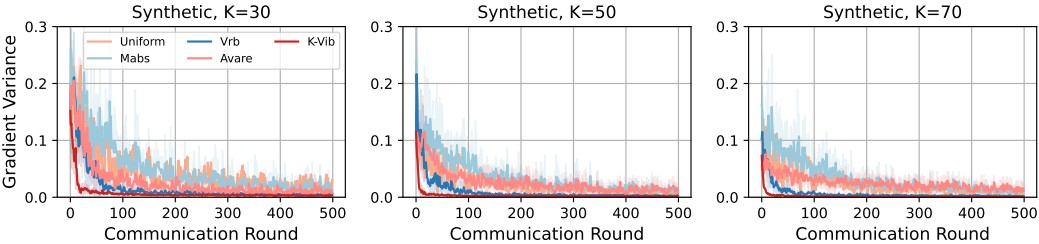

Figure 8: Gradient variance with different $K$

