# OpenReview forum: "Exploring Federated Optimization by Reducing Variance of Adaptive Unbiased Client Sampling"
_ICLR.cc/2024/Conference — ICLR 2024 Conference Withdrawn Submission_

### Official Review · Reviewer_iPwA · 2023-10-23

**Soundness:** 2 fair
**Presentation:** 1 poor
**Contribution:** 2 fair
**Rating:** 3
**Confidence:** 4

**Summary:**

The paper studies federated learning with client sampling. The authors consider the setting where we can choose the probabilities with which we want to sample clients and propose a new sampling procedure. Namely, they devise an objective that gives us unbiased estimates of the true regret. While the true regret requires information that we, in general, cannot access, the estimates, do not need it, and one can apply FTRL to the obtained problem to update the probabilities on the fly. The authors then proceed to a non-convex convergence guarantee for FedAvg with unbiased sampling. The last section of the paper is devoted to experiments on a synthetic dataset and FEMNIST.

**Strengths:**

1. The problem of client sampling can have a good impact on the FL literature if there is a significant empirical improvement.
2. The experiments clearly show an improvement obtained from the method on the problems where it was tested.

**Weaknesses:**

1. The theory is based on refining the previous results on the variance of clients and applying regret bounds. While there is nothing wrong with taking this approach, I wouldn't consider this a significant theoretical contribution and, therefore, I'd expect a thorough empirical evaluation.
2. The authors assume that all gradient norms are bounded. This effectively limits the data heterogeneity too. The claim that gradient clipping can be used to justify the assumption doesn't make much sense to me, since that would effectively change the method and require a different analysis.
3. The experiments are very limited and restricted to synthetic datasets and FEMNIST with a basic CNN.
4. It was quite hard for me to read the paper due to numerous typos (a subset of them listed below), inconsistent notation, and missing explanations. Sometimes authors refer to claims from later in the paper as if we've already seen them. Some parts of the proofs were quite unclear to me too (detailed in the Questions section of my review). I think the paper would benefit from a major revision.

## Minor
In line 5 of Algorithm 1, why is it $w^t$ and not $x^t$? This notation seems different from eq. (1) and line 9 of Algorithm 1. Similarly, in line 10, it is $w^t$ again
In line 8 of Algorithm 1, in the right-hand side it should be $x^{t,r-1}$
In line 8 of Algorithm 1, I think that $\xi_i$ should depend on $r$ as well, otherwise it's as if we are using the same sample
The notation $r$ for local iteration is not very good as $r$ typically denotes the round, which the authors denote by $t$
"as we concretely discussed in Appendix E.2" this sentece seems weird since we haven't seen Appendix E.2
"we are the first work that extends" -> "our work is the first to extend"
I can see different spellings of "K-VIB", for instance, in the last parargraph on page 2. Please make it consistent.
"To be consistency" -> "To be consistent"
"for simply notation" -> "for simplicty of notation"
"important sampling" -> "importance sampling"
"sampler hat outputs" -> "sampler that outputs"
"Thus, the above theorem would translate into regret guarantees". There has been no theorem up to that point in the text.
"Assumption 4 are" -> "Assumption 4 is"
"as we shown in Appendix 23" -> "as we show in Theorem 23"
and so on...

**Questions:**

1. In the proof of Theorem 8, why does the FTRL bound from Theorem 23 apply? Since in Line 16 of Algorithm 2 we sample $S^t$ according to $\tilde p^t$ rather than $p^t$, the feedback depends on $\tilde p^t$, so the whole procedure is not equivalent to FTRL.
2. In equation (41), the authors use unbiasedness of $\tilde \ell_t(p)$.
3. To apply FTRL regret guarantees, we must first show that the objective satisfies the assumptions of FTRL regret bounds. As far as I can see, objective (8) has unbounded gradients, so why can we apply the FTRL theory?
4. Why the probabilities given in equation (9) add up to 1? According to line 14 of Algorithm 2, $\tilde p_i^t = (1 - \theta) p_i^t + \theta \frac{K}{N}\$ for all $i=1,\dotsc, N$. As far as I can see, this means if we sum them, we get $\sum_{i=1}^N \tilde p_i^t = (1 - \theta)\sum_{i=1}^N p_i^t + \theta \frac{K}{N}\cdot N = 1 - \theta + \theta K = 1 + \theta (K - 1)$.

---

> ### Author Response · Authors · 2023-11-11
> **Response to reviewer#iPwA**
>
> Thanks for the detailed comments and notification provided by the reviewer. We would like to respond to the weaknesses and questions to support our paper.
>
> **Weaknesses**:
>
> - **"While there is nothing wrong with taking this approach, I wouldn't consider this a significant theoretical contribution and, therefore, I'd expect a thorough empirical evaluation."** The main contribution of this paper stands that "we are the first work that extends the independent sampling procedure on adaptive client sampling in federated optimization." The previous theory of regret bounds and independent sampling are studied on the non-convex stochastic optimization for data sampling. Differently, we are exploring adaptive unbiased client sampling in federated learning, which is a very different manner. We provide concrete theory and experiments to support our points in the paper. Can we have further suggestions from the reviewer about "thorough empirical evaluation"? We sincerely appreciate detailed comments that improve our work.
> - **"The authors assume that all gradient norms are bounded. "**   In the federated optimization literature (most of the federated optimization papers we have cited), plenty of works follow the bounded gradient assumption, while maintaining its powerful performance in applications. Using this assumption is not supposed to be the weakness of our work.  Besides, we involved the data heterogeneity in our analysis with the Assumption (11). Additionally, this paper mainly contributes an efficient and practical federated client sampler and proves its efficacy from theory and empirical study, instead of pursuing the state-of-art analysis in optimization.
> - **The experiments are very limited.** Our experiment setting follows previous works on federated optimization and client sampling studies. We believe the main points of the paper are fully covered by experiments. It will be appreciated if the reviewer provides direct requirements of the further experiment study. We are also willing to conduct further experiments if suggested.
> - **Typos and issues.** Thanks very much for your careful notification. We have modified all the minor issues and checked our paper carefully.
>
> **Questions**:
>
> - **"In the proof of Theorem 8, why does the FTRL bound from Theorem 23 apply?"** Please note that the term (B) in (41) has a similar form of FTRL objective in (34). Differently, we replace the true full feedback information with additional estimates according to sampled clients $S^t$ from $\tilde{p}^t$. As the additional estimates are unbiased as we discussed in Section 3.2, we can conduct the exact same analysis with Theorem 23 to prove the results by only replacing the $\ell_t(p)$ to $\tilde{\ell}_t(p)$. In conclusion, we can apply the FTRL bound from Theorem 23 in an expectation form.
> - **"In equation (41), the authors use unbiasedness of $\tilde{\ell}_t(p)$".** We are analyzing an expected regret bound with partial feedback constraints. Therefore, it is applicable.
> - **"Objective (8) has unbounded gradients, why can we apply the FTRL theory?"** Please note that objective (8) has a closed-form solution, which is analogous to Lemma 3 and the proof is detailed in Lemma 18. Therefore, we don't compute the gradients of objective (8). As we stated on page 6, the regularizer $\gamma$ of FTRL ensures that the distribution does not change too much and prevents assigning a vanishing probability to any clients.
> - **"Why the probabilities given in equation (9) add up to 1?"** Please note that we restricted that "**To be consistent, the sampling probability $p$ always satisfies the constraint $p_i^t > 0, \sum_{i=1}^N p_i^t = K, \forall t\in[T]$ in this paper**" in Definition 1. Therefore, we always have that $\sum_{i=1}^N \tilde{p}_i^t=(1-\theta) \sum_{i=1}^N p_i^t+\theta \frac{K}{N} \cdot N = K$, which consistent with our definition.
>
> In summary, we hope our response could better clarify our work. We are expecting the feedback from reviewers. Please feel free to comment us if further clarifications are required.

---

### Official Review · Reviewer_SHAe · 2023-10-27

**Soundness:** 2 fair
**Presentation:** 3 good
**Contribution:** 2 fair
**Rating:** 3
**Confidence:** 4

**Summary:**

This paper investigates federated learning with adaptive unbiased client sampling. The authors proposed a K-VIB sampler, which helps the FL algorithm to achieve better regret bound. The authors also proved the convergence of a federated learning algorithm, showing that the minimum of gradient converges to a neighborhood around the origin. Experiments are provided to bolster their theoretical contribution.

**Strengths:**

The authors considered minimizing the finite-sum objective, which is a standard but popular topic in FL. The paper is basically well-written. It is easy for readers to go through the paper.

The statements of theory are clear, with detailed proofs in the appendix. Numerical experiments are thorough, with several graphs to demonstrate the promising performance of the sampler.

**Weaknesses:**

Despite the strengths mentioned above, there are several weaknesses in the paper.

Weaknesses in contributions:

1. The theoretical part of the paper is extremely similar to the reference paper, Online Variance Reduction for Stochastic Optimization, by Zalán Borsos et al.. In fact, the design of K-VIB is essentially identical to the design of Variance Reducer Bandit (VRB), Algorithm 1 in Borsos's paper. It seems that the only difference is \sum p_i=1 being replaced by \sum p_i=K, thus resulting in the extra K-related term in the regret bound.  \sum p_i=K is also set in other published papers, like Nonconvex Variance Reduced Optimization with Arbitrary Sampling by Samuel Horváth and Peter Richtárik, so it does not count as a contribution of this paper, either. As for the benefit brought by independent sampling, nothing is mentioned except the smaller upper bound shown in Lemma 2. Therefore, it is hard to judge the advantage of using independent sampling. Therefore, I strongly suggest the authors to mention the benefit of independent sampling in details, it will be appreciated if the authors can compare the results between their paper and other similar papers where random sampling is used. After all, independent sampling is claimed to be one of your contributions.

2. The convergence analysis is flawed. The convergence result shown in the theorem is kind of weird: it is deterministic convergence.
In stochastic optimization (like here, where sampling is involved), convergence is usually in expectation, almost surely or with high probability. I don't think that it is possible to obtain a deterministic convergence result in a stochastic optimization algorithm. After all, the LHS of (12) is random, which depends on the samples, thus being different in each run of the algorithm. However, the first term in the RHS is deterministic. I also read the proof of the convergence analysis and found multiple severe errors. To name a few, starting from equation (43), the expectation is a mess. In equation (43), the expectation is taken conditioning on the sigma algebra until x^t (so here it should be a conditional expectation!). There should be only one expectation in the third term of the last line in page 25. Also, there is no need to take the second expectation in the equation (44), since the final result of (44) still depends on x^t. Also, in equation (52), the expectation is also conditional on x^t, so it is ridiculous to see, in equation (52),  the expectation disappears. Because of the omission of the conditional expectation, the authors manage to take average of both sides of (53) over time T. This is wrong, remember, equation (53) is conditional on the sigma algebra until x^t. You can not take summation with different conditions (e.g., x^t,x^{t-1}, etc.). The right step here is to take full expectation on both sides of (53) first, which removes all randomness, and them take summation over T. Only in this way, can you sum up equations for different T! Accordingly, the final result should be changed. What's more, showing the minimum of the norm of the gradient lies a neighborhood around zero is a too weak result. Given the current proof, the result can be strengthened without adding extra assumptions.

3. I am not able to check all proofs in details, so I strongly suggest the authors to go over every line of their proofs carefully. When expectation is taken, it is always a good idea to clarify the condition.

4. When claim that independent sampling is the optimal sampling technique, the reason should be independent sampling minimizes the upper bound of equation (14), instead of independent sampling minimizes the left term of equation (14). So please correct this statement in the appendix, the conclusion above equation (17).

Weakness in assumption:
The assumption f(x)−f(y)\leq F in the 13 is relatively strong. In fact, it can be replaced by f(x)\geq f_{inf}, that is, f(x) is lower bounded (which is a more standard assumption) without jeopardizing the convergence result. Since what you need in the convergence result is an upper bound in expectation, which can be satisfied if noticing that E(f(x^1)-f_{inf}) is always finite and E(f(x^T)-f_{inf}) is always positive.

Weaknesses in presentations:
There are many typos in the paper. To name a few, the local mini-batch SGD step in Algorithm seems to have a wrong superscript, T and R should be inputs of Algorithm 1. If the name of your sampler is K-VIB, please do not write K-Vib. The statement of Assumption 4 is also flawed.

**Questions:**

In Algorithm 1, g_i^t is denoted as the local update, but later g_i^t is called the gradient. It is confusing to me. Please clarify it.

When analyzing the regret bound, the weight \lambda_i is set as 1/N. How about general \lambda_i?

---

> ### Author Response · Authors · 2023-11-11
> **Response to reviewer#SHAe (1/2)**
>
> Thanks very much for your detailed comments! The following are our responses to the comments accordingly:
>
> **Weakneass in contributions**:
>
> - **Clarify theoretical contribution**.  First of all, we would like to state that the main contribution of this paper stands that "we are the first work that extends the independent sampling procedure on adaptive client sampling in federated optimization."  We mainly rely on the theoretical works of VRB and independent sampling proposed by Samuel Horváth and Peter Richtárik are studied on the non-convex stochastic optimization for data sampling. Differently, we are exploring adaptive unbiased client sampling in federated learning, which is a very different manner. In this paper, we first extend their theoretical results in federated optimization and present a new unbiased client sampling technique in federated learning.
> - **Comparison with other random sampling techniques**. Theoretically, the benefits of independent sampling have been given in Lemma 2, where we conclude that independent sampling could minimize the variance better than random sampling. Besides, we prove the convergence benefits of the K-Vib sampler (using independent sampling) are better than other adaptive samplers **MABS**, **Vrb**, and **Avare** (using random sampling) in Theorem 13 and remark paragraph.  Empirically, we conduct experiments in 4 settings to demonstrate the performance of the K-Vib sampler.  It would be appreciated if the reviewer pointed out other similar papers with random sampling that we may missed.
> - **Benefits of independent sampling**. The experiments show the superior performance of K-Vib over baselines. Besides, the K-Vib sampler partially inherits the theoretical design of **Vrb**, while innovatively replacing the sampling procedure with independent sampling. We believe the comparison between K-Vib and Vrb in Figure 4,5 could also indicate the conclusion in Lemma 2.
>
> For a brief summary, our work contributes greatly to the understanding of adaptive unbiased client sampling topics in the federated learning community. It supports the design of an efficient client sampler.
>
> - **The convergence analysis is flawed**. We humbly disagree with a few points in the comments. We have double-checked our analysis and are sure that the analysis has been fixed accurately. We split our response by points below:
>
>   **1. " In equation (43), the expectation is taken conditioning on the sigma algebra until x^t".**  **"There should be only one expectation in the third term of the last line in page 25."** In the last line of Equation (43), page 25, two expectation means different in $\mathbb{E}\left[\| \nabla f(x^t) - \mathbb{E}\left[d^t\right] \|^2\right]$. The first expectation is taking over time $t$ (condition on $x^t$. The second expectation is taken over sampling $S^t$, which reminds readers that the global estimates are built on an unbiased client sampling procedure. In Equation (44), we have removed all randomness.
>
>   **2. "In equation (52), the expectation is also conditional on x^t, so it is ridiculous to see, in equation (52), the expectation disappears."** We humbly highlight that the expectation has not disappeared, but is absorbed by sampling quality function $Q(S^t)$ as we defined in Equation (45) or (11). To reach that, we put a lot of effort into our analysis to remove randomness from Equation (44) to (49) and Equation (51) with mild assumptions. In this way, we gather all randomness in the sampling quality function $Q(S^t)$ in order to theoretically compare with other adaptive baselines in Theorem 13.
>
>   **3. "The right step here is to take full expectation on both sides of (53) first, which removes all randomness, and then take summation over T. Only in this way, can you sum up equations for different T! "** In the light of our response above, the Equation (53) can be summarised over time $T$.  We also note that we missed an expectation symbol on LHS of (12) or (53), which has been corrected.
>
>   **4. "showing the minimum of the norm of the gradient lies a neighborhood around zero is a too weak result."** The theoretical results stick to the standard analysis of non-convex optimization.  Our goal is to provide a convergence comparison between FedAvg with unbiased adaptive samplers as we stated in the remark of Theorem 13, instead of pursuing the state-of-art rate in federated optimization. Besides, the theoretical results also match the rate of $\mathcal{O}(1/\sqrt{T})$ in previous non-convex federated optimization as we described in Theorem 13.
>
> Given our response to the expectation problems, we are looking forward to your further comments on our theoretical analysis. We appreciate your comments helping us improve our work.

---

> > ### Author Response · Authors · 2023-11-11
> > **Response to reviewer#SHAe (2/2)**
> >
> > - **Expression issues**. We have fixed some issues as suggested: 1. replacing the assumption of f(x)-f(y); 2. making the conclusion above (17) accurate; 3. fixing typos and errors.
> >
> > - **g_i^t is denoted as the local update, but later g_i^t is called the gradient**:  g_i^t indicates the local update uploaded by clients. We have fixed the confusing content.
> >
> > - **How about general \lambda_i?** We set $\lambda_i = 1/N$ to save notations. If we need the regret bound with general $\lambda_i$,  we can replace the terms $1/N$ -> $\lambda_i$ or $1/N^2$ -> $\lambda_i^2$, which will not break the analysis. It also indicates that our method is compatible with reweighting algorithms related to $\lambda$.
> >
> > In all, we would like to express our gratitude for the constructive comments. We hope our response can resolve some concerns about our work. We are delighted to conduct further discussions or comments that may interest you.

---

> > ### Comment · Reviewer_SHAe · 2023-11-11
> >
> > Thanks for your reply. For the convergence analysis, I have some points to clarify based on your explanation.
> >
> > First, for 1., if the LHS expectation is taken over x^t, then on the RHS you cannot take an expectation over S^t. The randomness is different. The expectation should be taken for theoretical rigor, not to remind your readers. In equation 44, the randomness is removed based on x^t. Also, in my understanding, the unbiased sampling means that sampling is unbiased conditional on x^t, not S^t.
> >
> > Second, for 2., please remember that x^t is random, it is only deterministic if you condition on the sigma algebra of x^t. It means that f(x^t) is also random, and the randomness cannot be absorbed by Q(S^t), since S^t is generated after obtaining x^t.
> >
> > For 3., in (52), there is no expectation on the norm of gradient, but in (53), could you please explain the condition new expectation?

---

> ### Author Response · Authors · 2023-11-12
>
> Thanks for your reply. Here is my further explanation.
>
> First of all, $S^t$ is generated after obtaining $x^t$ means that we have $\mathbb{E}\left[d^t\right] = \sum_{i=1}^N \lambda_i g_i^t$ about sampling $p^t$ probability for the first equation of (44). To my opinion, the term $\mathbb{E}_{t} \left[\left\| \nabla f(x^t) - \mathbb{E} \left[d^t\right] \right\|^2\right] = \mathbb{E}_t \left[\left\| \nabla f(x^t) - d^t\right\|^2\right]$ can means the same. According to the comments, we have modified the term to the last one to be rigorous.
>
> From (52) to (53), we should take full expectation on both (52) or (53) before summation as suggested. We have fixed the point in the new version.
>
> For a brief summary, we are very thankful to the reviewer for pointing out the flaw in the analysis, which we have fixed accordingly. Most importantly, the main conclusion in Theorem 13 is not broken. Therefore, the main contribution of this paper still holds.
>
> Given that, we sincerely request the reviewer reassess our paper according to our contribution. And, we would like to know if there are any other improvements that we can make to reach the top conference paper.

---

### Official Review · Reviewer_6zGx · 2023-11-03

**Soundness:** 3 good
**Presentation:** 2 fair
**Contribution:** 3 good
**Rating:** 5
**Confidence:** 3

**Summary:**

In this study, the Partial Participation regime in Federated Learning is the primary focus. In this regime, only a subset of all clients can participate in the communication round due to the large number of clients and communication bottleneck. Optimal sampling strategies are under consideration; however, obtaining optimal probabilities requires collecting information from all clients. To relax this condition, the adaptive sampling method, K-Vib, is proposed. Theoretical analysis of the proposed sampler is conducted from an online optimization perspective, leading to the acquisition of convergence guarantees for the FedAvg method with this novel sampling procedure.

**Strengths:**

I appreciate the writing style of this text. The work demonstrates a solid structure, and the motivation is well articulated. I appreciate that the objective function is presented in a general form. The introduction section includes a well-structured literature review, and I particularly like that the authors have provided additional review material in the appendix section. This approach allows the main ideas to be presented in the primary text while offering supplementary details for reference.

I also commend the authors for highlighting specific modifications of the FedAvg algorithm in the text. However, I recommend avoiding color highlighting, as it can pose difficulties for color-blind individuals. The structured contribution section effectively explains the results.

It's worth noting that all definitions, lemmas, and theorems are well formulated. I reviewed the appendix, and the proofs appear to be accurate. Nevertheless, I'm not an expert in online optimization, so there might be nuances I've missed.

The experiments serve to illustrate the theoretical findings, but it might be beneficial to expand and provide a more extensive discussion in this section.

**Weaknesses:**

**Abstract**
In my view, the abstract is somewhat concise and lacks clarity. There are some confusing phrases that need further explanation:
>A line of 'free' adaptive client sampling

Could you please provide clarification on the meaning of 'free' in this context?

>Promising sampling probability

This phrase is also unclear. Could you elaborate on what is meant by "promising sampling probability"?

>which solves an online convex optimization respecting client sampling in federated optimization"

This sentence is confusing as it's not evident how one can solve "optimization," and the repeated use of the term "optimization" in close proximity may lead to misunderstandings. Please provide more context or clarification regarding this statement.

**Introduction**
The paper presents a robust literature review. Nevertheless, it lacks coverage of studies that explore cyclic client participation using the random reshuffling method introduced in the following papers.

Cho, Y. J., Sharma, P., Joshi, G., Xu, Z., Kale, S., & Zhang, T. (2023). On the convergence of federated averaging with cyclic client participation. arXiv preprint arXiv:2302.03109.

Malinovsky, G., Horváth, S., Burlachenko, K., & Richtárik, P. (2023). Federated learning with regularized client participation. arXiv preprint arXiv:2302.03662.

Additionally, it would be beneficial to compare the proposed approach with the method utilizing arbitrary sampling for accelerated Federated Learning, as introduced in the following paper:

Grudzień, M., Malinovsky, G., & Richtárik, P. (2023). Improving Accelerated Federated Learning with Compression and Importance Sampling. arXiv preprint arXiv:2306.03240.

In the pseudocode of Algorithm 1, the authors have employed object-oriented-style notation, such as "sample.p" and "sample.update." This notation can be confusing and appears somewhat incongruous alongside mathematical formulas. I recommend revising the pseudocode for clarity and coherence.

**Preliminaries**

>The second term represents the bias induced by the multiple local SGD steps in federated optimization to save communication (McMahan et al., 2017).

This bias is called client drift and this tern was introduced the paper and became standard in federated learning. I suggest to also use this term for clarity.

Karimireddy, S. P., Kale, S., Mohri, M., Reddi, S., Stich, S., & Suresh, A. T. (2020, November). Scaffold: Stochastic controlled averaging for federated learning. In International conference on machine learning (pp. 5132-5143). PMLR.

**Methodology of K-Vib Sampler**

>Assumption 4 (Lipschitz gradient). Each objective $f_i(x)$ for all $i \in[N]$ is G-Lipschitz, inducing that for all $\forall x, y \in \mathbb{R}^d$, it holds $\left\|\nabla f_i(y)\right\| \leq G$.

It is unclear why this assumption is termed "Lipschitz gradient" since it necessitates the objective functions to be Lipschitz, which implies that the gradient is bounded, but not Lipschitz. Could you please clarify this aspect?

>Assumption 5 (Local convergence).

Could you please provide further elaboration on this assumption, along with examples to illustrate how it applies to specific cases?

>Besides, the Lipschitz gradient also can be justified by using gradient clipping during the practical optimization of deep learning models to prevent exploding gradients and guarantee differential privacy (Kairouz et al., 2021).

While clipping can ensure that the assumption of bounded gradients is met, it's important to note that the clipping operator is not included in the proposed algorithm. Therefore, I believe this argument does not apply in this context.

>Assumption 9 (Smothness). Each objective $f_i(x)$ for all $i \in[N]$ is $L$-smooth, inducing that for all $\forall x, y \in \mathbb{R}^d$, it holds $\left\Vert\nabla f_i(x)-\nabla f_i(y)\right\Vert \leq L\Vert x-y\Vert$.

I can recommend to consider more general assumption with personalized constants $L_i$: $\left\Vert\nabla f_i(x)-\nabla f_i(y)\right\Vert \leq L_i\Vert x-y\Vert$

>Assumption 11 (Bounded Global Variance). We assume the weight-averaged global variance is bounded, i.e., $\sum_{i=1}^N \lambda_i\left\Vert\nabla f_i(x)-\nabla f(x)\right\Vert^2 \leq \sigma_g^2$ for all $x \in \mathbb{R}^d$.

The current inequality employs a universal constant for the bound, which is limiting in its generality. I recommend adopting a more general assumption where the heterogeneity bound is proportional to $\Vert x \Vert$ and a certain constant:

$ \sum_{i =1}^{N} \lambda_i \left\Vert\nabla f_i(x)-\nabla f(x)\right\Vert^2 \leq B\Vert\nabla f(x)\Vert^2+\zeta^2 \quad \forall x \in \mathbb{R}^d$

Such assumption is used in the following papers:

Gorbunov, E., Horváth, S., Richtárik, P., & Gidel, G. (2022). Variance reduction is an antidote to byzantines: Better rates, weaker assumptions and communication compression as a cherry on the top. arXiv preprint arXiv:2206.00529.

Karimireddy, S. P., Kale, S., Mohri, M., Reddi, S., Stich, S., & Suresh, A. T. (2020, November). Scaffold: Stochastic controlled averaging for federated learning. In International conference on machine learning (pp. 5132-5143). PMLR.

**Experiments**

The current plots are relatively small, making it challenging to discern the behaviors of the methods, particularly in the middle plot of Figure 4. Additionally, the current plots do not employ distinct markers, which poses difficulties for individuals with color blindness. Could you please adjust the plots accordingly?

**Questions:**

Please check questions in the Weaknesses section.

I would also like to inquire if it is possible to obtain convergence guarantees for both strongly convex and general convex cases or if results can be derived for non-convex functions that satisfy the Polyak-Lojasiewicz condition.

In general, I find this work to be acceptable; however, there are some clarity issues and a lack of convex analysis that concern me. I would be willing to consider raising the score if these issues are addressed. My current score is 5.

---

> ### Author Response · Authors · 2023-11-11
> **Response to reviewer#6zGx**
>
> We really appreciate the supportive and construction comments! We would like to address the concerns and weaknesses of our paper with the following points.
>
> **Abstract**:
>
> - **A line of 'free' adaptive client sampling**:  We define a federated client sampling technique as free if it is orthogonal to conventional federated learning protocol (e.g., fedavg) and doesn't require additional local computation and communication.
> - **Promising sampling probability**:  The "promising" means that the sampling probability utilized by the server could enhance federated optimization.
> - **Modification**: "solves an online convex optimization (problem) respecting client sampling in federated optimization" -> "which solves an online convex optimization problem respecting federated client samplings".
>
> We have modified our abstract with more precise expressions accordingly.
>
> **Introduction**:
>
> - **Literature review**: we have referred to the suggested works.
> - **Comparision with work**: We refer to the contribution claim from the paper[Grudzień, M., Malinovsky, G., & Richtárik, P. (2023)] "We finally present a complete method for Federated Learning that incorporates all necessary ingredients: Local Training, Compression, and Partial Participation." to clarify our differences. The benefits of the paper are obtained from the novel incorporation of different techniques. In contrast, we assume the clients have no additional computation for other techniques and we focus on the sampling techniques and global estimates for model updating while maintaining the largest compatibility to secure aggregation and re-weighting algorithms. Additionally, our sampling technique could also work with communication compression algorithms, such as quantization and sparsification.
> - **Pseudocode**: we have modified the pseudocode accordingly.
>
> **Prelinimaries**:
>
> - **Local drift**: we have modified the term for clarity.
> - **Lipschitz gradient**: this is a writing fault. We mixed up the Assumptions 4 and 9 in our draft. We have modified assumption 4 to "bounded gradients".
> - **Local convergence**: this assumption supports us in the analysis of our sampling method in a successful optimization procedure. It indicates that our samplers work in a converging federated learning procedure that with proper hyper-parameters of learning rates, batch size, and local steps settled. Therefore, we assume the norm of local updates decays over communication rounds in federated learning.
> - **Bounded gradient**: in practical cases, companies deploying federated learning algorithms will incorporate the optimization algorithm with privacy-preserving techniques (we refer to the URL below). As a result, gradient clipping is an optional trick for our method in both practice and theory. URL: https://machinelearning.apple.com/research/differential-secrecy. Besides, since the bounded gradient is widely used in the cited federated optimization literature, we hope our bounded gradient be accepted. We are not contributing a new optimization analysis but an efficient client sampler in this paper.
> - **Smoothness and global variance**: we will modify our analysis to a more general version with suggested assumptions. Besides, we want to clarify that our assumption partially inherits from widely accepted work in federated optimization[1], and our analysis is to prove the present sampler is effecant in federated learning.
>
> [1]Sashank J. Reddi, Zachary Charles, Manzil Zaheer, Zachary Garrett, Keith Rush, Jakub Konecný, Sanjiv Kumar, Hugh Brendan McMahan: Adaptive Federated Optimization. ICLR 2021.
>
> **Experiements**:
>
> - **Figures**: Thanks very much for the notification. We will revise the marker of our figures to make it clear.
>
> **Questions**:
>
> - **Convex analysis**: it is applicable to obtain the convergence guarantees for both strongly convex and general convex cases. We can replace Equation (43) with corresponding assumptions and extend the proof. We would like to note that our experiments on Synthetic datasets cover convex cases from a practical perspective.
>
> In summary, we greatly appreciate the constructive comments to improve our work. We have modified some of the weak points and updated the PDF. For the overall analysis with different assumptions, we will work on and update it in the next version. Finally, please let us know if we addressed your concerns about our work. We are looking forward to your feedback and are delighted to answer any questions that may come up.

---

### Author Response · Authors · 2023-11-16
**Summary of the revision.**

We have refined and revised our paper according to the comments. The revision can be summarised as follows:

Theorem:
- **Bouned gradient assumption**: we removed the bounded gradient assumption in our analysis and refined the description of the local convergence assumption accordingly.
- **Convergence analysis**: we refined the convergence analysis, where we fixed flaws (expectation problem, typos, notation errors) and removed bounded gradient assumption. We believe we have addressed the weaknesses in our theoretical analysis.

Importantly, **we emphasize that our main conclusion remains unchanged.**

Writing:
- **Minor**: we carefully checked the notation and minor issues. We have corrected most of them.
- **Color of figures**: we modified the color of the figures to ensure they are clear. Due to page limitations, we are unable to adjust the size of the figures.

Finally, we appreciate the constructive comments provided by the reviewers. We are looking forward to further comments from reviewers.